# Continuously trapped matter-wave interferometry in magic Floquet-Bloch band structures

Xiao Chai [1], Eber Nolasco-Martinez [1,3], Xuanwei Liang [1,3], Jeremy L. Tanlimco [1,3], E. Quinn Simmons[1], Eric Zhu [1], Roshan Sajjad[1], Hector Mas[1], S. Nicole Halawani [1], Alec Cao [1,2] & David M. Weld [1] ✉

Trapped matter-wave interferometry offers the promise of compact high-precision local force sensing. However, noise in the trap itself can introduce new systematic errors which are absent in traditional free-fall interferometers. We describe and demonstrate an intrinsically noise-tolerant Floquet-engineered platform for continuously trapped atom interferometry. A non-interacting degenerate quantum gas undergoes position-space Bloch oscillations through an amplitude-modulated optical lattice, whose resulting Floquet-Bloch band structure includes Landau-Zener beamsplitters and Bragg mirrors, forming the components of a Mach-Zehnder interferometric force sensor. We identify, realize, and experimentally characterize magic band structures, analogous to the magic wavelengths employed in optical lattice clocks, for which the interferometric phase is insensitive to lattice intensity noise. We leverage the intrinsic programmability of the Floquet band synthesis approach to demonstrate a variety of interferometer structures, highlighting the potential of this technique for quantum force sensors which are tunable, compact, simple, and robust.

Interferometry has been a tool of discovery in physics for centuries[1,2]. While originally demonstrated with photons, the wave-particle duality[3] of matter enables the realization of interferometers in electrons[4], neutrons[5], atoms[6], and molecules[7]. The interferometric sensitivity is proportional to spacetime area enclosed by the interferometer loop and thus scales quadratically with free-fall time for untrapped particles. This has driven the precision frontier towards large-scale experiments featuring 100-meter-scale drop towers[8] or even low Earth orbit[9]. In contrast, continuously-trapped interferometers can reach very large spacetime areas in the Earth's gravity field without requiring long free-fall time, large experimental size, or spaceflight, and offer the practical advantages of compactness and truly local sensing[10–18]. However, instabilities in the trapping potential result in dephasing. While this can be partly mitigated by resonant optical mode filtering[19] or differential measurement schemes[17], such dephasing still represents a primary limitation of trapped matter-wave interferometers.

In this work we describe and experimentally characterize a new and highly flexible class of atom interferometer based on position-space Bloch oscillations[20,21] of trapped non-interacting Bose-condensed lithium through loops in a Floquet-engineered optical lattice band structure[22–25]. The sensitivity of this interferometer scales inversely with the applied force, making it especially suitable for detection of weak forces. Instead of using traditional Raman or Bragg pulses, matter-wave splitting and recombination is realized via tunable Landau-Zener transitions at quasimomentum-selective Floquet-Bloch band crossings. This makes the interferometer intrinsically insensitive to fluctuations in initial momentum, pulse duration, and laser phase.

[1]Department of Physics, University of California, Santa Barbara, CA, USA. [2]Present address: JILA, University of Colorado and National Institute of Standards and Technology, and Department of Physics, University of Colorado, Boulder, CO, USA. [3]These authors contributed equally: Eber Nolasco-Martinez, Xuanwei Liang, Jeremy L. Tanlimco. ✉e-mail: weld@physics.ucsb.edu

Immunity against trap intensity fluctuations is achieved by the use of an infinite family of magic Floquet-Bloch band structures, which generalize the concepts of magic wavelengths and magic lattice depths[26]. We experimentally verify that magic Floquet-Bloch bands exhibit first-order insensitivity to lattice amplitude noise. These advantages are a consequence of the remarkable degree of design flexibility offered by Floquet band engineering, which takes inspiration from related concepts in condensed matter physics[27,28] and photonics[29,30] and enables the synthesis of a wide variety of interferometer loops with different characteristics and tunable force response.

## Results

### Working principle

The central component of the interferometers we describe is a horizontal one-dimensional optical lattice amplitude-modulated to hybridize the Bloch bands (Fig. 1). The lattice potential takes the form $V(x,t) = -(V_0 + \delta V \sin\omega t)\cos^2(k_L x)$, where $V_0$ is the static lattice depth, $\delta V$ and $\omega$ are the modulation depth and frequency respectively, $k_L = 2\pi/\lambda$ is the lattice laser wavenumber, and $\lambda = 1064$ nm is the lattice

laser wavelength. This yields a spatially and temporally periodic single-particle Hamiltonian to which Floquet-Bloch theory can be readily applied[22]. Just as quasimomentum $q$ arises from the breaking of continuous to discrete space translation symmetry, the analogous breaking of continuous to discrete time translation symmetry gives rise to quasienergy bands, which are periodic in $\hbar\omega$, where $\hbar$ is the reduced Planck constant. As shown in Fig. 1c, the modulation opens up gaps between Floquet-Bloch quasienergy bands where they intersect, at quasimomentum $q = q_r$. For band indices $n$ and $n'$ (henceforth $S$, $P$, $D$, etc., following orbital notation) of the two bare Bloch bands $E_{n,q}$, $E_{n',q}$, the resonance condition is $E_{n,q_r} - E_{n',q_r} = \hbar\omega$. When a matter wave in one band traverses the avoided crossing at $q = q_r$, it coherently splits into a superposition state between the two output bands[31–33]. Tuning the modulation depth $\delta V$ controls the gap, and thus the Landau-Zener transition probability $\mathcal{P}$ can be adjusted to 50%[34]. Arbitrary synthesis of the RF modulation waveform allows fully programmable manipulation of both the location and strength[34] of all such avoided crossings between quasienergy bands: these will become the beamsplitters and other control elements comprising the interferometer.

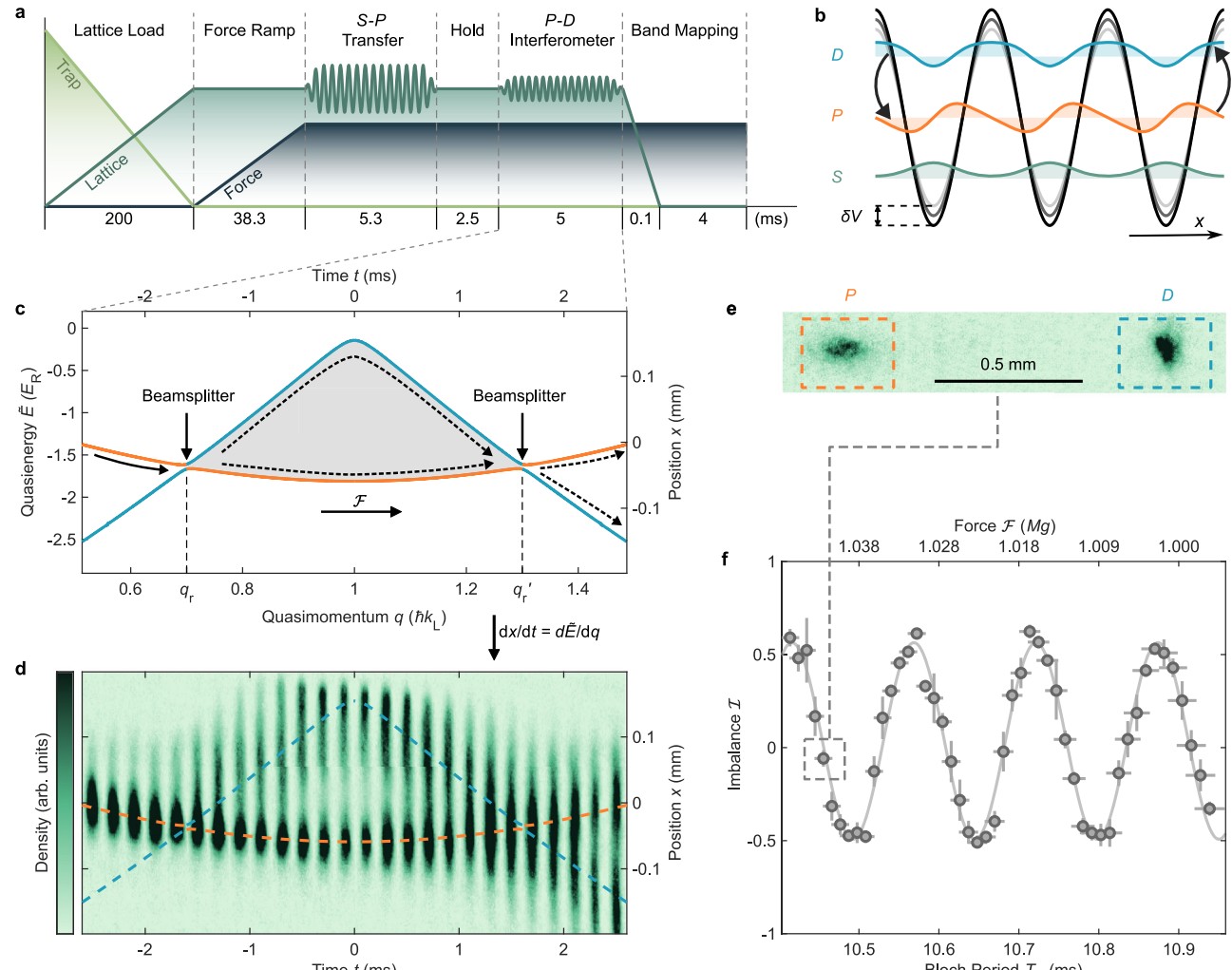

**Fig. 1 | Floquet-Bloch matter-wave interferometry. a** Experimental sequence. **b** Resonant amplitude modulation of the optical lattice hybridizes the $P$ and $D$ (second and third) Bloch bands. **c** Interferometric loop formed by the Floquet-Bloch band structure and Landau-Zener beamsplitters. A matter wave Bloch oscillates through the paths indicated by the arrows in response to a force $\mathcal{F}$. The color of the Floquet-Bloch bands corresponds to the Bloch states in (**b**) with maximal wavefunction overlap. **d** Time sequence of *in situ* images of a condensate traversing the interferometric loop. The space-time trajectories map out the

Floquet-Bloch band structure. **e** Band-mapping spatially separates the two output port populations after a 4 ms time of flight. **f** Output population imbalance $\mathcal{I}$ versus applied force. $Mg$ is the mass times the local gravitational acceleration, used here only as a force scale; the lattice is horizontal. In all figures, vertical error bars are the standard error of 3 repeated measurements and horizontal error bars are the estimated uncertainties from calibration fits (Supplementary Information section 4.2).

In the presence of a force $\mathcal{F}$ along the lattice direction, the acceleration theorem[35] guarantees that the quasimomentum $q$ of a matter wave evolves as $q(t) = q(0) + \mathcal{F}t$, with the Bloch period defined by traversal of the entire Brillouin zone:

$$T_{\mathrm{B}} = \frac{2\hbar k_{\mathrm{L}}}{\mathcal{F}}. \qquad (1)$$

Crucially, this Stückelberg-type evolution in quasienergy and quasimomentum[36,37] also results in motion in real space: the mean position $x(t)$ of a wavepacket in a Floquet-Bloch band with quasienergy dispersion relation $\widetilde{E}(q)$ evolves according to the group velocity as $dx/dt = d\widetilde{E}/dq$[21]. The displacement associated with such position-space Bloch trajectories can be especially large for light atoms like lithium; separations on the millimeter scale are straightforwardly attained. This means that by adjusting the modulation waveform to synthesize a particular Floquet-Bloch band structure between two Landau-Zener avoided crossings, we are synthesizing a particular spacetime trajectory for the atoms in the interferometer loop: the quasienergy band dispersion is simply a scaled image of the center-of-mass time evolution[23], as shown in Fig. 1c, d. Although the two wavepackets experience different dispersions, any differential spreading experienced during the loop traversal will be reversed when they recombine. Components of the atomic wavefunction which are separated at an initial Landau-Zener beamsplitter and traverse the two arms of such a loop accumulate a relative dynamical phase which depends on the applied force. Recombination of the matter waves at the second beamsplitter completes a Landau-Zener-Stückelberg-Majorana interferometer[38,39]. The differential phase acquired by each arm $\phi_{\mathrm{Int}}$ manifests as a population imbalance between the two output bands $\mathcal{I} = p_{\mathrm{L}} - p_{\mathrm{U}} \approx C \cos \phi_{\mathrm{Int}}$ by the adiabatic-impulse approximation[40], where $p_{\mathrm{L(U)}}$ denotes population fraction in the lower (upper) Floquet-Bloch output band. The contrast $C$ is maximized by 50-50 beamsplitting fraction $\mathcal{P} = 0.5$. The interferometer phase is $\phi_{\mathrm{Int}} = \phi_{\mathrm{Dyn}} + 2\phi_{\mathrm{Sto}}$, where $\phi_{\mathrm{Sto}} \in [-\pi/2, -\pi/4]$ is the Stokes phase imparted by one of the Landau-Zener transitions, which depends only weakly on the applied force. $\phi_{\mathrm{Dyn}}$ is the differential dynamical phase

$$\phi_{\mathrm{Dyn}} = \frac{T_{\mathrm{B}}}{2\hbar^2 k_{\mathrm{L}}} \int_{q_r}^{q_r' = q_r + \Delta q} dq (\widetilde{E}_{\mathrm{U},q} - \widetilde{E}_{\mathrm{L},q}), \qquad (2)$$

where $\widetilde{E}_{\mathrm{L(U)},q}$ denotes the quasienergy. The dynamical phase is proportional to $T_{\mathrm{B}}$ and thus $1/\mathcal{F}$; the force sensitivity scales linearly with the energy-momentum area enclosed by the interferometer loop.

## Experimental implementation

The experimental apparatus with which we realize this interferometer builds upon previous work[23]. A $^{7}$Li Bose-Einstein condensate (BEC) is prepared in the $|f, m_f\rangle = |1, 1\rangle$ magnetically sensitive state within a crossed dipole trap. To eliminate interaction-induced dephasing[17], we apply a magnetic field of 543.6 G to tune the s-wave scattering length to zero[41] using lithium's shallow Feshbach zero-crossing. The atoms are then adiabatically loaded into the ground band of a horizontal one-dimensional optical lattice (Fig. 1a) with a lattice depth $V_0 = 8.45 E_{\mathrm{R}}$, where $E_{\mathrm{R}} = \hbar^2 k_{\mathrm{L}}^2/2M$ is the recoil energy, and $M$ is the atomic mass. Subsequently, we ramp up a magnetic field gradient effecting a nearly uniform force $\mathcal{F}$ along the lattice direction to initiate Bloch oscillations. A strong 120 kHz lattice amplitude modulation pulse is then applied to transfer the atoms from the $S$ to $P$ band with 100% Landau-Zener probability. This initial transfer allows us to implement the interferometer using $P$-$D$ hybridized Floquet-Bloch bands, the lowest combination which supports magic band structures. After a 2.5 ms hold time, a 127.438 kHz modulation is ramped up within 100 μs to load the atoms adiabatically into the upper $P$-$D$ hybridized Floquet-Bloch band. The modulation depth is calibrated to be $\delta V = 0.35 E_{\mathrm{R}}$ for

50–50 beam splitting (Supplementary Information section 4.3). The modulation is sustained for 4.8 ms, during which time the atoms traverse the Brillouin zone edge and complete the interferometer loop at the second crossing. The second crossing is created by the same modulation and therefore is mirrored around the zone edge from the first crossing; this allows the interferometer loop to close. Figure 1d shows a series of snapshots capturing the atoms traversing the interferometer loop, where the center-of-mass motion clearly maps out the Floquet-Bloch band structure[21,23]. Finally, the modulation is ramped down in 100 μs to convert atoms in upper (lower) Floquet-Bloch bands back into the static $P$ ($D$) Bloch bands. We perform band-mapping[42] and absorption imaging to read out the band populations with high fidelity (Fig. 1e). Figure 1f demonstrates a functioning interferometer, displaying a measured interference fringe in which the population imbalance depends sinusoidally on the inverse of the applied force $\mathcal{F}$. See Methods for details.

## Magic band structure

In addition to force, the differential dynamical phase in general also depends sensitively on lattice depth. If this were always the case it would spoil the performance of force sensors of this type[43]. Fortunately, the flexibility and large available parameter space of multi-frequency Floquet band engineering allows for the design and creation of magic band structures that null out the lattice depth dependence. The condition for magicness of a Floquet-Bloch band loop is $\partial\phi_{\mathrm{Int}}/\partial V_0 = 0$. This condition cannot be realized in any simple loop whose constituent bands include the ground ($S$) band. The reason for this is that the perturbative effect of nonzero $V_0$ on the free-particle parabolic dispersion serves to repel the different bands; since the $S$ band uniquely only experiences repulsion from above, its energy monotonically decreases with increasing lattice depth (and more steeply than any other band), precluding the existence of any nonzero $V_0$ for which the energy difference with another band has zero derivative[26]. Thus, the simplest pair of bands that can realize a "magic-depth" interferometer are the $P$ and $D$ bands. For example, as demonstrated numerically in Fig. 2a, the magic condition is realized for a $P$-$D$ interferometric loop extending from $q_r = 0.8\,\hbar k_{\mathrm{L}} \rightarrow q_r' = 1.2\,\hbar k_{\mathrm{L}}$ at lattice depth $V_0^{\mathrm{M}} \approx 8.85\,E_{\mathrm{R}}$. This approach can be generalized to different loop areas and interferometer geometries: such magic band structures can be found for essentially any quasimomentum range (Supplementary Information section 2) and any set of constituent excited bands.

To test the predicted magic band structure, we first measure the population imbalance at fixed force and modulation waveform across a range of lattice depths near this value. The results, shown in Fig. 2b, indeed indicate a broad region with $\partial\phi_{\mathrm{Int}}/\partial V_0 \approx 0$ centered on $V_0 = 8.85\,E_{\mathrm{R}}$. To probe directly the performance of magic versus non-magic band structures, we measure force scans for three closely spaced lattice depths away from the magic condition (Fig. 2c) and for three closely spaced lattice depths centered at the magic condition (Fig. 2d). Measurements far from the magic condition exhibit sensitive dependence on lattice depth and greater fluctuations in imbalance for a given lattice depth; in contrast, force measurements at or near the magic condition are quiet and consistent across a range of lattice depths. These results clearly demonstrate both the successful experimental implementation of magic Floquet-Bloch band synthesis and its utility for force sensing.

## Force response

One key motivation for continuously trapped atom interferometry is the ability, in principle unlimited by geometrical constraints, to scale up the spacetime area of the interferometer loop to increase sensitivity. To probe such scaling, we use the intrinsic flexibility of Floquet-synthesized interferometry to create a sequence of interferometers with increasing spacetime loop area. Specifically, we measure interference fringes produced using loops which subtend spans of

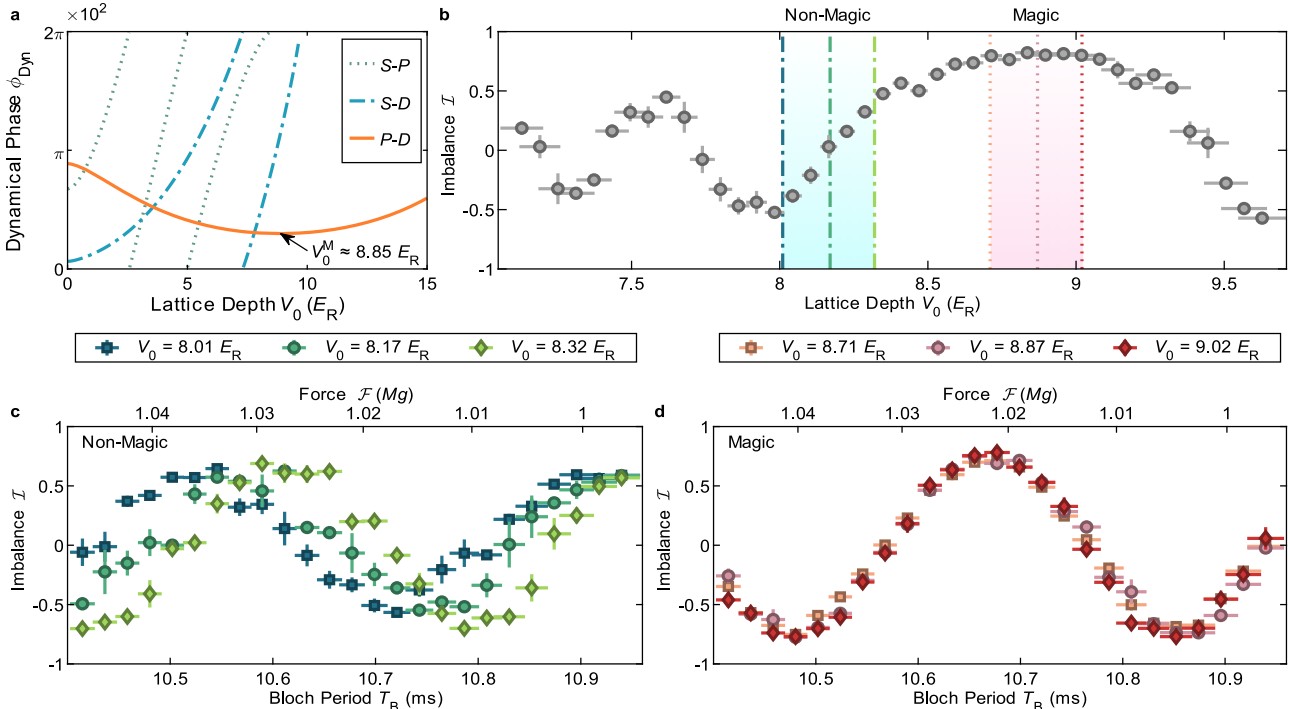

**Fig. 2 | Magic band structures. a** Numerically calculated dynamical phase (Eq. (2)) modulo $2\pi \times 10^2$ for S-P, S-D, and P-D interferometric loops (with 100 kHz, 190 kHz, and 143 kHz drives, respectively) as a function of lattice depth for $T_B = 10.7$ ms estimated with static band energies; only the P-D loop exhibits a magic condition at nonzero $V_0^M \approx 8.85\,E_R$. **b** Output population imbalance at $T_B = 10.70$ ms as a function of lattice depth. Vertical dashed (dotted) lines indicate the lattice depths chosen for force measurements far from (close to) the magic condition. **c, d** Imbalance as a function of applied force away from (**c**) and close to (**d**) the magic condition at the aforementioned lattice depths.

quasimomenta $\Delta q = q_r' - q_r$ ranging from $0.2\,\hbar k_L$ to $1\,\hbar k_L$, keeping the interferometer loop symmetric about the Brillouin zone edge. This probes the performance of interferometers with different momentum-space beamsplitter separations. For each loop size, we recalculate the magic condition and adjust the lattice depth accordingly. At the magic condition, the range of acceptable lattice depth fluctuations decreases with increasing loop size (Fig. 3b), but for all measurements remains comfortably above the experimentally achievable lattice depth stability. As shown in Fig. 3a, the measured interference fringes become finer as the loop area increases, indicating an increased force response. The most likely candidate for the reduction of fringe contrast at higher $\Delta q$ is an inhomogeneous transverse field gradient, causing imperfect loop closure (Supplementary Information section 6). Figure 3c shows the inverse of the measured fringe period, which is proportional to the force sensitivity, as measured by sinusoidal fits for each interferometer loop. The results match the analytical theory without any fitting parameters. These data clearly demonstrate a well-understood path to scaling up the force sensitivity in continuously-trapped interferometers by increasing loop area.

## Programmability

Beyond this straightforward increase in beamsplitter separation, the inherent programmability and power of Floquet band engineering also offer a wide variety of other methods to enhance or tune the performance of trapped matter-wave interferometers. Possibilities include the use of switchable or pulsed beamsplitters, the inclusion of higher bands beyond D, multifrequency Floquet band engineering, and coherent control of the beamsplitting phase. The final set of experiments we describe explores and demonstrates all these capabilities, stocking a versatile toolbox for programmable trapped matter-wave interferometry.

In the initial such experiment, we address a key limitation of the Floquet band engineering approach to matter-wave interferometry:

any modulation frequency which resonantly couples two bands (say, P and D) at a particular quasimomentum will in general also resonantly couple to other bands at different quasimomenta. Such couplings give rise to undesired "leaks" to other states, resulting in parasitic interferometers and reduced contrast. Undesired couplings thus can severely limit the design space for Floquet band interferometers, and become unavoidable for larger-area loops. The problem is illustrated in Fig. 4a, which shows the dressed energy band structure (Supplementary Information section 1.6) of an interferometer subtending about half a Brillouin zone in quasimomentum. The modulation frequency used to create the beamsplitters by coupling the P and D bands at about 0.5 and 1.5 $\hbar k_L$ also resonantly couples both the D and F bands and the P and F bands, at different values of quasimomenta within the interferometer loop. While not shown, higher bands beyond F are also coupled in this range via multiphoton transitions. The modulation scheme in Fig. 4a shows the simplest way around this problem: smoothly turn on and off the modulation so that the bands are only coupled during the beamsplitting operation. Fig. 4b shows a fringe measured in this rather large-area interferometer using pulsed beamsplitters, demonstrating that switchable couplings can be implemented without compromising interferometer performance. This greatly opens up the possible design space for dressed-band interferometry, for instance by allowing loops to extend over multiple Brillouin zones.

Of course, coupling to additional higher bands is not always undesirable; in fact it can be used to design interferometers with enhanced loop area and force sensitivity. Fully flexible use of this capability requires the ability to program arbitrary interband coupling strengths and quasimomenta, which in turn requires the simultaneous use of multiple modulation frequencies. Fig. 4c shows the quasienergy band structure (calculated using many-mode Floquet theory[44]) of an experiment demonstrating this capability. We modify a loop with $\Delta q = 0.6\,\hbar k_L$ by applying an additional strong modulation at 196 kHz to

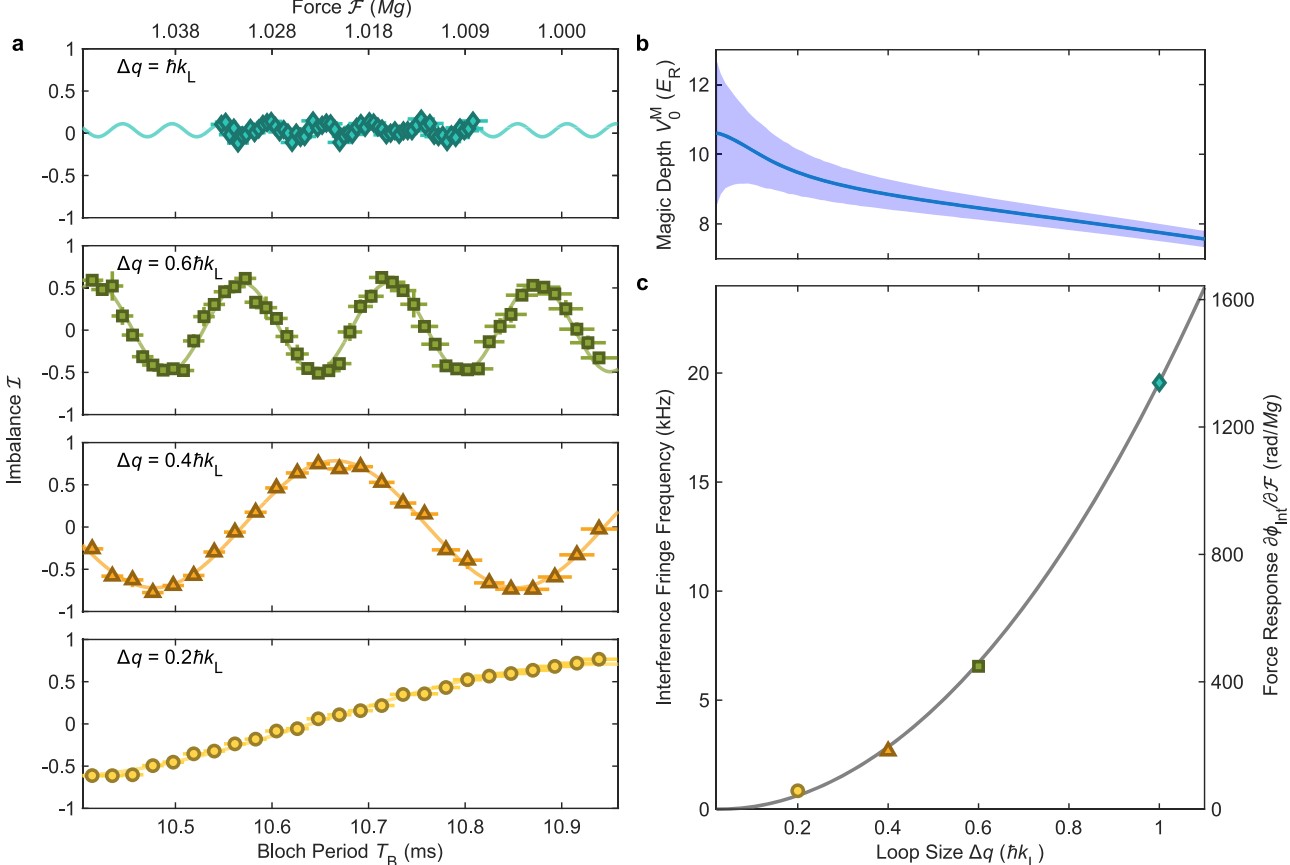

**Fig. 3 | Tuning the interferometer response. a** Interference fringes for varying loop sizes $\Delta q = q'_r - q_r$, each at the respective magic condition. Solid lines are sinusoidal fits. **b** Theoretically calculated magic depth as a function of loop size. Boundaries of the shaded area mark the lattice depth where the interferometer phase deviates by $\pi/4$. **c** Interferometric response as a function of loop size as defined by the fringe frequency. Data points represent fit results from (**a**), and the solid line is the theoretical prediction obtained from the fit-parameter-free analytical theory. The corresponding force response is calculated at $\mathcal{F} = Mg$.

hybridize the $P$ and $F$ bands at quasimomenta $0.85\,\hbar k_L$ and $1.15\,\hbar k_L$, with approximately 100% Landau-Zener transition probability. The loop which includes the $F$ band has a larger area and should be more sensitive to the applied force than the original $P$-$D$ loop. Measured interference fringes for the two interferometers, shown in Fig. 4d, demonstrate the predicted decrease in fringe period and increase in sensitivity. This result demonstrates that arbitrary multi-frequency modulations can be used to construct tailored band structures to control and optimize loop geometry and response of trapped matter-wave interferometers.

Finally, to optimize the response of an interferometer to arbitrary perturbations it is helpful to be able to tune the overall phase of the interference fringes. This is straightforwardly accomplished for trapped Floquet-band matter-wave interferometers by adjusting the phase of the lattice modulation waveform which creates one of the beamsplitters. Fig. 4e diagrams such an experiment in a $P$-$D$ interferometer with $\Delta q = 0.4\,\hbar k_L$, which builds on the separated-pulse interferometry demonstrated in Fig. 4a, b with the addition of a phase shift between the two beamsplitter modulation pulses. Fig. 4f shows the results: for three different overall forces, scanning the phase of the second beamsplitter pulse shifts the fringe by up to $2\pi$. This demonstration of coherent control of beamsplitter phase is useful for determining the contrast when the force cannot be easily scanned, and enables straightforward biasing of interferometer response to the point of maximum sensitivity. A relative phase sensitivity of $1.01 \times 10^{-4}$ over a 20 min measurement is estimated from the data shown in Fig. 4f (Supplementary Information section 9). Moreover, this method serves as a practical tool for characterizing interferometer stability for a fixed set

of parameters. Supplementary Fig. S7 shows that the phase-scan fringes are nearly immune to 10% variations in initial quasimomenta and pulse durations, highlighting the intrinsic robustness inherited from utilizing Landau-Zener transitions for beamsplitters, as achieving the desired transition probability (and thus the maximal contrast) only depends on the pulse overlapping the crossing at some point rather than its exact duration.

## Discussion

We have described and experimentally demonstrated a simple, versatile, and extensible class of trapped matter-wave interferometer constructed from magic Floquet-Bloch band structures in an amplitude-modulated optical lattice. Key virtues of this approach include compactness, flexibility, high ultimate sensitivity unlimited by device size, and intrinsic robustness against trap-induced dephasing and pulse errors. Compact trapped interferometers suitable for weak force measurement may find application in fifth force searches or similar probes of physics beyond the standard model[45–48] as a complementary approach to free-fall atom interferometry.

While our numerical simulations reveal that the force response scales up when the interferometric loop encompasses multiple Bloch oscillations (Supplementary Information section 8), the ultimate limits on the sensitivity of this technique remain to be explored. Although in the experiments we present the coherence was limited by technical imperfections like transverse field gradient inhomogeneity, this can be straightforwardly improved. Possible avenues for such improvements include using higher-order magic band structures which cancel additional systematics, magnetically

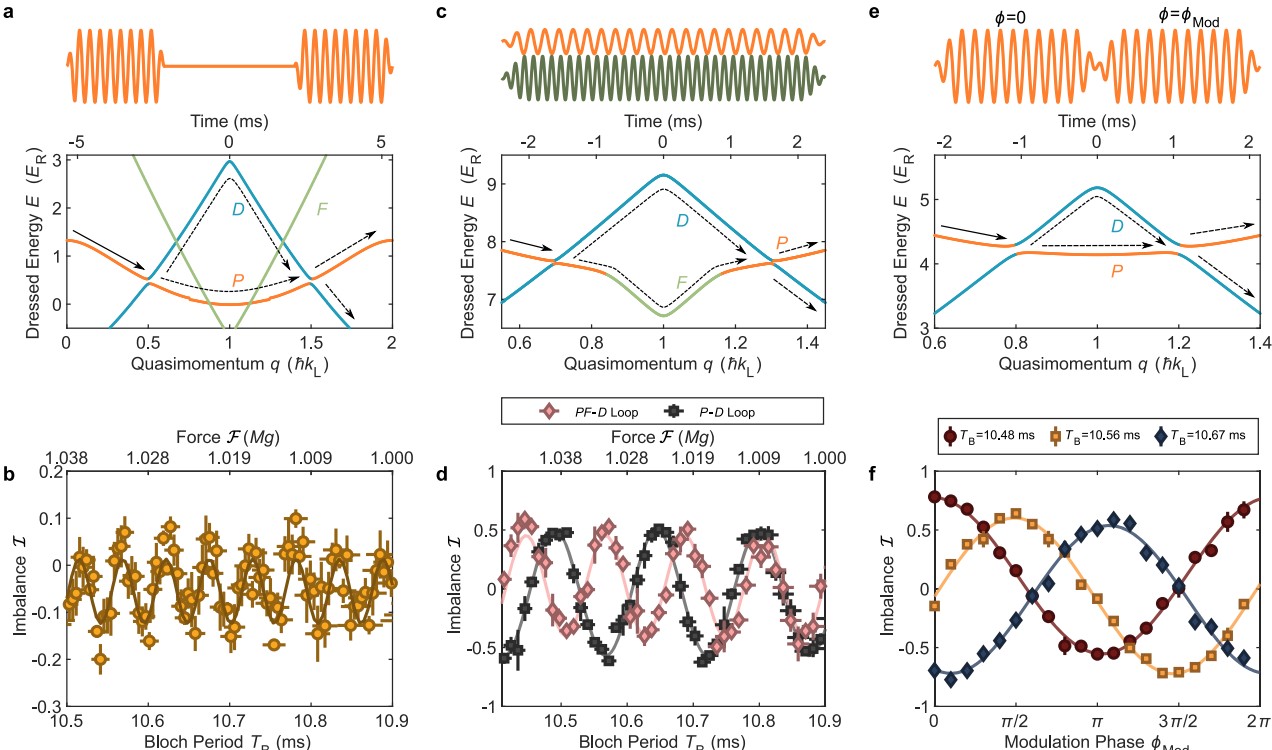

**Fig. 4 | Programmable Floquet synthesis of various interferometer structures. a** Dressed energies of the $\Delta q = \hbar k_L$ loop and the corresponding modulation waveform. The modulation is pulsed to avoid coupling with the $F$ band. Recall that quasimomentum evolves via the acceleration theorem as $q(t) = q(0) + 2\hbar k_L(t/T_B)$, so the time to traverse an interferometer loop of fixed quasimomentum separation is proportional to $T_B$. **b** Interferometric fringe generated by the temporally separated pulses. **c** Dressed energies and modulation waveform of the $\Delta q = 0.6\,\hbar k_L$ loop in the presence of a second 196 kHz drive (dark green), which hybridizes the $P$ and $F$ bands. **d** Interferometric fringes with and without the second drive. **e** Dressed energies and modulation waveform of the $\Delta q = 0.4\,\hbar k_L$ interferometer loop. The phase of the second pulse is shifted by $\phi_{Mod}$ relative to the first. **f** Output population imbalance as a function of $\phi_{Mod}$ for three different forces.

insensitive states or isotopes with intrinsically weak interactions[49], implementing improved field control, and adding a resonant low-finesse cavity as a mode cleaner. Past work has shown[23] that space-time trajectories can be fully controlled to enable much larger loops, holding at large separations, and executing multiple Bloch oscillations; this means that loop area can in principle grow almost without bound, so any improvements in coherence will directly enhance force sensitivity. The flexibility, power, and large design parameter space offered by the Floquet engineering framework, along with the excellent match between interferometer performance and both numerical and analytical theory, should allow the use of optimal control and machine learning techniques[50] to design more complex interferometer sequences for enhanced robustness and sensitivity. While the low-mass isotope ⁷Li is helpful here for enabling large spatial separation, the technique could be expanded to heavier atoms by using higher-band transitions to generate higher momentum transfer. Finally, given the band-synthesis approach used here, it may be instructive for future developments to compare to, and draw inspiration from, related efforts in driven condensed matter[27,28].

# Methods
## Experimental details
Our experiments begin with a BEC of $2 \times 10^5$ ⁷Li atoms produced in a horizontal crossed optical dipole trap with trapping frequencies $(\nu_x, \nu_y, \nu_z) = (151.9, 185.0, 239.4)$ Hz. The atoms are prepared in the $|f = 1, m_f = 1\rangle$ ground state, with the magnetic field initially held at 715 G along the vertical $z$ direction to ensure a positive scattering length. The condensate fraction is greater than 95%, having evaporated in the trap until the thermal components are no longer visible. The field is then

ramped down over 1 s to 543.6 G, where the $s$-wave scattering length vanishes. Subsequently, the crossed dipole trap is switched off within 200 ms, while a one-dimensional optical lattice with 110 μm waist is simultaneously ramped up (Fig. 1a). This loads the atoms into the ground ($S$) band of the optical lattice with a narrow quasimomentum spread of ~ 0.13 $\hbar k_L$ (full width at half maximum). Since the lattice is oriented horizontally along the $x$-axis, gravity does not produce a force along the lattice, as is the case in ref. 51. Instead, a field gradient $\partial B_z/\partial x$ generated by the coils is ramped up over 38.3 ms, which applies a force and initiates Bloch oscillations. The lattice depth is modulated by varying the radio-frequency power sent to an acousto-optic modulator.

## S-P transfer
To perform $P$-$D$ atom interferometry with magic band structures, the atoms must first be prepared in the $P$ band. This is accomplished via Floquet modulation. After the 38.3 ms field gradient ramp, the atoms acquire a momentum such that the central quasimomentum reaches $q = -\hbar k_L$. A strong lattice amplitude modulation with $\delta V = 1.68\,E_R$ is then ramped up within 100 μs opening up a $S$-$P$ hybridized gap at $q \sim 0.5\,\hbar k_L$. As the matter wave moves towards the Brillouin zone center and traverses the gap, it adiabatically follows the instantaneous Floquet state that connects to the $P$ Bloch band. The modulation is subsequently ramped down within 100 μs, resulting in a near-unity transfer efficiency to the $P$ band. We chose to couple at this quasi-momentum to perform our transfer because the perturbation coupling strength $\langle\varphi_{S,q}|\cos^2(k_L x)|\varphi_{P,q}\rangle$ is nonexistent at $q = 0$ and $q = \hbar k_L$ due to the parity selection rules. For this same reason, we use the $S$ and $D$ bands for calibration of lattice depth (Supplementary Information section 4.1) and do not perform a population inversion at the center of our symmetric $P$-$D$ interferometer.

## Data acquisition and analysis

After band mapping with a 4 ms time of flight (Fig. 1e), the interferometer output appears as spatially separated atomic clouds corresponding to the $P$ and $D$ bands. We perform standard absorption imaging to obtain the atomic column density. The imaging beam is aligned along the $z$-axis, and the 5 μs imaging pulse is applied while the magnetic field remains on. To mitigate imaging imperfections, we apply a least-squares regression fringe removal procedure. The $P$ and $D$ band fractions are then calculated by integrating the atom numbers within regions of interest as indicated in Fig. 1e.

## Data availability

Due to the large volume of the dataset, the data used in this manuscript are available from the corresponding author upon request.

## Code availability

The code for calculating band structures and magic conditions is available on Code Ocean.

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

## Acknowledgements

We thank Naceur Gaaloul, Rui Li, and Matthew Glaysher for useful discussion; Samyuktha Ramanan for experimental assistance; and Yifei Bai for physical insight and a critical reading of the manuscript. We acknowledge support from the Army Research Office (W911NF-22-1-0098 and W911NF-20-1-0294), the Noyce Foundation and the Eddleman Quantum Institute, and from the NSF QLCI program through Grant No. OMA-2016245. E.N.-M. acknowledges support from the UCSB NSF Quantum Foundry through the Q-AMASEi program (Grant No. DMR-1906325). S.N.H. acknowledges support from the NSF NRT program under grant 2152201. A.C. acknowledges support from the NSF Graduate Research Fellowship Program (Grant No. DGE2040434).

## Author contributions

X.C., E.N.-M., X.L., J.L.T., E.Z., and S.N.H. ran the experiment and performed the measurements. X.C. and X.L. implemented lattice and magnetic field stabilization. E.N.-M. performed field curvature cancellation. E.N.-M., J.L.T., E.Q.S., R.S., H.M., and A.C. carried out early experimental explorations. X.C., E.N.-M., X.L., J.L.T., and S.N.H. analyzed the data. X.C. conceptualized the analytical model. X.C., X.L., E.Q.S., E.Z., and A.C. performed numerical calculations. D.M.W. developed the idea for the experiment and supervised the work. X.C., E.N.-M., X.L., J.L.T., E.Z., S.N.H. and D.M.W. wrote the manuscript. All authors contributed to the discussion and interpretation of the results.

## Competing interests

The authors declare no competing interests.
