## [Transparent Peer Review file · Nature Communications]

Continuously trapped matter-wave interferometry in magic Floquet-Bloch band structures

Corresponding Author: Professor David Weld

Version 0:

Reviewer comments:

Reviewer #1

(Remarks to the Author)

The authors demonstrate Landau–Zener–Stueckelberg interferometry using Floquet–Bloch bands that are engineered through amplitude modulation of an external periodic potential.

This work is remarkable because the authors succeed in splitting a wavepacket in quasimomentum space and recombining it, showing coherence even after the two atomic components have spatially separated by about half a millimeter.

Most importantly, they identify a range of lattice depths for which the Stueckelberg phase difference accumulated by the interferometer is stationary with respect to lattice depth.

They refer to this configuration as “magic.”

The interferometer is operated for different initial and final quasimomenta, and the authors demonstrate a variable phase sensitivity to an external uniform force.

The experimental results are well described, and the paper is clearly written.

The experimental achievements are significant, and I am convinced that this work could stimulate new ideas within the community and advance the state of the art in trapped-atom interferometry.

This is likely sufficient to warrant publication in Nature Communications.

However, I have a few reservations regarding the applicability of this scheme to real precision measurements, given the approach presented in the paper.

Below, I explain my main concerns.

If I understand correctly, the scheme links the force sensitivity to the ability to measure the Bloch oscillation period with high precision—similarly to what is done with standard Bloch oscillations of matter waves in periodic potentials.

However, whereas in conventional Bloch oscillations the final atomic momentum must be precisely measured with respect to the initial zero-momentum state (which may fluctuate), the present scheme cleverly measures a phase difference proportional to the Bloch period, with a proportionality factor given by the energy difference between two bands integrated over a quasimomentum interval.

On one hand, the authors identify a magic configuration where this proportionality factor is stable against intensity fluctuations.

On the other hand, I suppose that it cannot be determined with high absolute accuracy, as it depends on the absolute lattice intensity, which is difficult to measure precisely.

This, in my view, strongly limits the applicability of the proposed device to precision metrology.

Another concern relates to the spatial resolution of force measurements.

As the authors emphasize, a longitudinal harmonic confinement prevents the two split wavepackets from recombining at the critical final quasimomentum at the same time, thus reducing the interference contrast.

For this reason, any attempt to measure a spatially varying force would likely face the same limitation.

For these two reasons, I conclude that, although I consider this work a remarkable piece of physics and a source of inspiration for future developments in the field, the technique still lacks a clear path toward practical applications in precision sensing.

Below I list a few additional comments that I would like the authors to address:

The authors claim that the interferometer is noise-tolerant.

Can they quantify how sensitive the splitting and recombination processes are to lattice position fluctuations?

Moreover, I imagine that the interferometric phase remains affected by unwanted lattice accelerations.

The final imbalance is said to be insensitive to fluctuations of V_0 .
What about fluctuations in δV , i.e., the modulation amplitude of the lattice?

Considering my previous comments, what kind of measurement could realistically be performed with this interferometer to go beyond the current state of the art?

An estimate of the achievable sensitivity and spatial resolution is missing.
Could the authors provide quantitative estimates of the force sensitivity and its dependence on spatial resolution?

Another potential source of contrast reduction could be different dispersions of the two wavepackets.
Is this effect relevant, or do the two wavepackets perfectly match in width at the moment of the second beam splitter?

I am wondering how condensed the atomic cloud used in the experiment is.
Could the interferometric scheme also work with a thermal sample? Does the interferometer have a white-light-like character?

Considering that lithium is a very light species, what would significantly change if this scheme were implemented using heavier atoms, in terms of optical power and final sensitivity?

(Remarks on code availability)

Reviewer #2

(Remarks to the Author)

The manuscript describes the theory and experimental implementation of an optical lattice atom interferometer that uses amplitude modulation of the optical lattice to perform the atom optics operations: beam splitter, mirror, recombination. The operations are achieved by transferring Li atoms between the p and d bands of the optical lattice, taking advantage of "magic trap depths", specific laser intensities where the interferometer is shown to be first order insensitive to lattice amplitude noise. The scheme presented is in the same vein as several similar implementations using frequency modulation by the JILA and U Washington teams, but different in important ways.

It remains to be seen whether experimental sensitivity and systematic suppression can reach level competitive with the current state of the art. Despite this, the results are an important step forward for lattice atom interferometer and highlight a new type of optical lattice atom interferometer. The experimental data is compelling and interpretation is physically sound. The manuscript is well written, and the data is nicely presented.

General comments:

1. The interferometer scheme leads to an unusual interferometer phase dependence on the parameter of interest, i.e it is proportional to $1/\text{force}$, instead of the usual proportionality to force. This means that keeping the force small is important for achieving a large phase and therefore good sensitivity. This is acknowledged by the authors in the conclusion, "The force sensitivity in the weak-force regime improves as $1/F$, a feature that could be exploited in more generality by introducing an accelerated lattice to realize a frame transformation and cancel part of the force." Of course, the problem is that the main advantage of the interferometer, ability to achieve long measurement times, disappears, since the lattice has to be traveling at g to cancel out the force. I wonder if this can be commented on.
2. The sensitivity of the interferometer seems to be far from competitive for precision measurements. However, this is an important consideration on whether it would be useful for applications. The interferometer is phase stable, but a short discussion of possible systematic effects could help the reader assess possible applications. This is particularly the case given the complex residual effects from interactions with the optical lattice.
3. "The most likely candidate for the reduction of fringe contrast at higher is an inhomogeneous transverse field gradient, causing imperfect loop closure" - Using atoms in a magnetically sensitive state makes the experiment quite sensitive to environmental fields. Are collisions impossible to deal with in a magnetically insensitive state?

Specific comments:

1. "The interferometric sensitivity is set by the spacetime area enclosed by the interferometer loop and thus scales quadratically with freefall time for untrapped particles."
- I suggest replacing "set by" with "proportional", since sensitivity also depends on measurement rate – duty cycle and number of atoms per cycle.
2. "without requiring mode filtering with a resonant cavity." Cavity wouldn't filter the amplitude noise unless the noise is at very high bandwidth, beyond the resonance linewidth of the cavity. This seems unlikely since most laser intensity noise is in the acoustic band.
3. In figure 2b, it would make sense to convert from imbalance to phase. That would make the assessment of intensity insensitivity more accurate, which is now dependent on being on the side or extrema of the fringe.

(Remarks on code availability)

I took a quick look but I did not run the code.

(Remarks to the Author)

In this work, Chai et al. present a flexible method of precision measurement for measuring extremely weak applied forces. This method of interferometry is described through the lens of Bloch oscillations via Floquet physics and Landau-Zener transition probabilities. Unlike many other types of interferometers, this new class of interferometer is extremely insensitive to many external factors and can be achieved in trap. In reviewing this paper, I was impressed by how thorough the authors were in their description of the new class of interferometer and was struck by the cleanliness of the data presented in the paper. In a time when new methods for quantum sensing are highly sought after, I find this work to be well-timed and highly beneficial to the field of precision measurements. **I highly recommend this work for publication in Nature Communications**, given the above comments and the relevant nature of the work to a growing field of quantum science and technology. I firmly believe that this work, along with the Supplemental Information, will be a relevant guide to those who would like to implement similar experimental methods.

In reviewing the manuscript, we formulated several questions and comments that we would like the authors to respond to and/or provide small changes to the manuscript and supplemental information.

1. One general overarching note is that, while the Supplementary Information is very thorough, it seemed to be that the supplementary information was needed in certain parts of the main manuscript to understand what was going on. The authors may want to consider small changes to the main manuscript to make sure that the Supplementary Information is providing additional information but is not needed for one to understand what is going on in the main text.
2. In Figure 1e, it would be nice to note the imaging time-of-flight time and to provide a scale on the image to give an idea for the relative size of the signals and their separation in the experiment. The time-of-flight should also be included in the relevant part of the methods section.
3. On the 3rd page, first column, in the sentence that starts with "This phase is measured by interfering..." (lines 119-127), the authors define the upper and lower populations, but then in the ideal contrast equation, ρ is not defined. I understand that this is defined in the Supplementary information, however, this parameter should at least be named in the main text, e.g. "... where ρ is the Landau-Zener probability (see Supplementary Information section 1)." The authors could then move the reference to Supplementary Information to the end of this sentence, rather than the start of the sentence. The main reason I found this to be confusing is that the same letter was used for population fractions ρ_U and ρ_L , so having a parameter that is ρ , sans subscript, led me to become confused that maybe this was another population fraction term.
4. The authors calibrated the modulation amplitude δV only at the magic condition (Supp Mat section 4.3). How sensitive are the results to small deviations in δV across the lattice depths used in the experiment? When plotting the imbalance as a function of lattice depth (e.g. in Fig. 2b), it might be helpful for the authors to provide a brief comment on how possible uncertainties in δV could affect these measurements.
5. In a literature review for similar work using Floquet-state engineering for interferometry, we found the following paper relevant to this work: *T. Rodzinka et al, Nature Comm 15, 10281 (2024)*. While this paper does not take away from the novelty of the current work under review, I found that it may be important to cite this work, perhaps in the Methods section, to make it clear that this work stands alone from other work previously done with these types of methods. For example, "Since the lattice is oriented horizontally along the x -axis, gravity does not produce a force along the lattice, as is the case in Ref.~\cite{Rodzinka_2024}." Of course, I can't recall if the limit on number of references for Nature Comm is 50, but considering the similar Bloch oscillation method used in that paper, we thought it would be good to mention prior to publication.
6. We are assuming that the "shallow Feshbach zero-crossing" for Lithium is sufficiently shallow that when the magnetic gradient is applied, that the change in scattering length along the extent of the BEC induces negligible interaction-induced dephasing. Is this a correct assumption of the experimental setup? In the main manuscript and in the supplemental information, I did not get a sense of the relative strength of the magnetic field gradient used in the experiment. In section 5 of the supplemental information, under \textbf{Gradient Coil Current}, it may be nice for the authors to give a range for the gradients using in the work, in relatable units of G/cm.
7. In Figure 4, the x -axes are given in terms of the Bloch period $T_{\text{B}} \sim \text{ms}$ or the quasimomentum $q \sim (\hbar k_L)$. Because these two quantities are directly related through the equation $q(t) = q(0) + \mathcal{F}t$, the figure could be made clearer if the authors added a small annotation or a visual cue to remind the readers of this mapping.
8. In the Supplementary Information, lines 62-64 state, "We adopt a coarse-graining argument [3], which states that an infinite number of avoided crossings below a certain scale can be ignored because of the finite experimental time scale." In this work, how did the authors choose the scale for the neglected avoided crossings? How is the threshold quantitatively justified or verified in their analysis?
9. In Eq. (S43) of the Supplementary Information, the authors approximate the change in Stokes phase as a function of the change in lattice depth V_0 is very small compared to the change in the Dynamical phase over the same range. It may be relevant to make a reference to both Eq. (S23) and Figure S2 preceding or following this approximation.

(Remarks on code availability)

Reviewer #4

(Remarks to the Author)

(Remarks on code availability)

Version 1:

Reviewer comments:

Reviewer #1

(Remarks to the Author)

The authors have addressed all the questions raised during the first review with extensive and clear explanations. I confirm that the work is remarkable and merits publication in Nature Communications.

(Remarks on code availability)

Reviewer #2

(Remarks to the Author)

I thank the authors for the replies. I suggest adding the measured phase sensitivity of 1 part in 10^4 in 20 minutes to the manuscript. I know it's not competitive with the state of the art, which may make atom fountain dinosaurs uncomfortable... however, let's remember that this is a relatively new scheme that is just now seriously being explored.

Otherwise, it is my belief that this paper should be published without much delay. The quality of the data and analysis is excellent and the authors were quite careful with their precision claims. We need to become a lot more open to new approaches if this field is to advance in the future.

(Remarks on code availability)

Reviewer #3

(Remarks to the Author)

The authors have satisfactorily addressed all concerns/points raised in our review and have made agreeable changes to the manuscript. We have no further comments and recommend the paper for publication without further revision.

(Remarks on code availability)

Reviewer #4

(Remarks to the Author)

(Remarks on code availability)

Dear Editors and Referees,

We first would like to thank the referees for their very positive feedback, including statements such as: *“The experimental achievements are significant; This is likely sufficient to warrant publication in Nature Communications; The results are an important step forward for lattice atom interferometer and highlight a new type of optical lattice atom interferometer; The experimental data is compelling and interpretation is physically sound; I highly recommend this work for publication in Nature Communications, given the above comments and the relevant nature of the work to a growing field of quantum science and technology.”*

We are grateful to the referees for their insightful and constructive comments, which have significantly helped us improve the clarity and quality of our work. We found the feedback on the discussion of the sensitivity and systematics particularly valuable. We have addressed all points raised by the referees and revised the manuscript accordingly. Our response to each comment is outlined in detail in the attached document which itemizes the changes made to the manuscript and Supplementary Information, including the addition of new figures and enhanced theoretical discussion.

We believe the revised manuscript now presents a stronger and clearer account of our experimental achievements in creating a magic Floquet-Bloch interferometer. We look forward to hearing from you regarding the resubmission.

Sincerely,
All authors

1 Response to reviewer 1

General Comments

The authors demonstrate Landau–Zener–Stueckelberg interferometry using Floquet–Bloch bands that are engineered through amplitude modulation of an external periodic potential. This work is remarkable because the authors succeed in splitting a wavepacket in quasimomentum space and recombining it, showing coherence even after the two atomic components have spatially separated by about half a millimeter. Most importantly, they identify a range of lattice depths for which the Stueckelberg phase difference accumulated by the interferometer is stationary with respect to lattice depth. They refer to this configuration as “magic.” The interferometer is operated for different initial and final quasimomenta, and the authors demonstrate a variable phase sensitivity to an external uniform force. The experimental results are well described, and the paper is clearly written. The experimental achievements are significant, and I am convinced that this work could stimulate new ideas within the community and advance the state of the art in trapped-atom interferometry. This is likely sufficient to warrant publication in Nature Communications.

Response: We thank the reviewer for their thoughtful comments and for their judgement that the experimental achievement we report is significant and could stimulate new ideas and advance the state of the art. We appreciate the conclusion that the results likely warrant publication in Nature Communications. We respond below point-by-point to the referee’s specific comments and suggestions, and have edited the manuscript in response.

Comment 1.1

However, I have a few reservations regarding the applicability of this scheme to real precision measurements, given the approach presented in the paper. Below, I explain my main concerns. If I understand correctly, the scheme links the force sensitivity to the ability to measure the Bloch oscillation period with high precision—similarly to what is done with standard Bloch oscillations of matter waves in periodic potentials. However, whereas in conventional Bloch oscillations the final atomic momentum must be precisely measured with respect to the initial zero-momentum state (which may fluctuate), the present scheme cleverly measures a phase difference proportional to the Bloch period, with a proportionality factor given by the energy difference between two bands integrated over a quasimomentum interval. On one hand, the authors identify a magic configuration where this proportionality factor is stable against intensity fluctuations. On the other hand, I suppose that it cannot be determined with high absolute accuracy, as it depends on the absolute lattice intensity, which is difficult to measure precisely. This, in my view, strongly limits the applicability of the proposed device to precision metrology.

Response: We thank the reviewer for the valuable discussion about whether precision measurements can be performed through our approach. We agree that the interferometer phase in our scheme is directly related to the band structure, hence the lattice depth. However, our magic band structure technique does ensure that the force (acceleration) measurement is not vulnerable to the uncertainty in lattice depth at first order. We address this point in detail in the following response and have added relevant discussion to the Supplementary Information. To evaluate the measurement uncertainty, we rewrite the interferometer phase as (ignoring the Stokes phase and assuming perfect

beamsplitting):

$$\phi_{\text{Int}} = \frac{1}{\hbar \mathcal{F}} \int_{q_r}^{q_r' = q_r + \Delta q} dq \left(\tilde{E}_{\text{U},q} - \tilde{E}_{\text{L},q} \right) = \frac{\hbar}{\mathcal{F}} \mathcal{A}(V_0, \omega), \quad (\text{R1})$$

where

$$\mathcal{A}(V_0, \omega) = \int_{k_r(V_0, \omega)}^{k_r'(V_0, \omega) = k_r + \Delta k} dk \left[\tilde{\Omega}_{\text{U},k}(V_0) - \tilde{\Omega}_{\text{L},k}(V_0) \right] \quad (\text{R2})$$

denotes the integrated band energy difference controlled by the lattice depth V_0 and the modulation frequency ω ; the quasimomentum and quasienergy are normalized by \hbar :

$$k_r = \frac{q_r}{\hbar}, \quad \Delta k = \frac{\Delta q}{\hbar}, \quad \tilde{\Omega}_{\text{U},k} = \frac{\tilde{E}_{\text{U},\hbar k}}{\hbar}, \quad \tilde{\Omega}_{\text{L},k} = \frac{\tilde{E}_{\text{L},\hbar k}}{\hbar}. \quad (\text{R3})$$

Suppose we determine the absolute phase ϕ_{Int} from interference fringes with an uncertainty $\sigma_{\phi_{\text{Int}}}$, the acceleration a can then be determined as

$$a = \frac{\mathcal{F}}{M} = \frac{\hbar}{M} \frac{\mathcal{A}(V_0, \omega)}{\phi_{\text{Int}}}, \quad (\text{R4})$$

with the uncertainty

$$\frac{\sigma_a}{a} = \sqrt{\left(\frac{\sigma_{\phi_{\text{Int}}}}{\phi_{\text{Int}}} \right)^2 + \left(\frac{\sigma_{\mathcal{A}}}{\mathcal{A}} \right)^2 + \left(\frac{\sigma_{\hbar/M}}{\hbar/M} \right)^2}. \quad (\text{R5})$$

Since the ratio \hbar/M can be measured to a precision far beyond other sources of errors, we will ignore the contribution of $\sigma_{\hbar/M}$. The factor $\mathcal{A}(V_0, \omega)$ is proportional to the lattice wave vector k_{L} and the recoil frequency E_{R}/\hbar , both of which can be determined with high precision. In addition, it also depends on the control parameters ω and V_0 . ω is an RF frequency which can be determined very precisely. While V_0 is in general less precisely known, the contribution due to uncertainty in V_0 is reduced by the magic band structure. Near the magic condition where $\partial \mathcal{A} / \partial V_0 |_{V_0 = V_0^{\text{M}}} = 0$, the uncertainty σ_{V_0} only has a second-order contribution to $\sigma_{\mathcal{A}}$. Therefore, we can rewrite Eq. (R5) as

$$\frac{\sigma_a}{a} \approx \sqrt{\left(\frac{\sigma_{\phi_{\text{Int}}}}{\phi_{\text{Int}}} \right)^2 + \left(\frac{\sigma_{V_0}}{V_0^{\text{T}}} \right)^4}, \quad (\text{R6})$$

where

$$V_0^{\text{T}} = \sqrt{2 \left| \frac{\mathcal{A}}{\partial^2 \mathcal{A} / \partial V_0^2 |_{V_0 = V_0^{\text{M}}}} \right|} \quad (\text{R7})$$

characterizes the tolerance against lattice depth uncertainty. Eq. (R6) tells us the requirement for lattice depth uncertainty is not stringent if one wants to demonstrate precision measurements. The underlying idea is similar to the magic wavelength approach in optical lattice clocks, where the first-order differential light-shift from the lattice is canceled, and the effects of lattice intensity fluctuation are minimized.

To provide a sense of how precisely we need to determine V_0 , we plot V_0^{T} as a function of loop size in Fig. R1. We notice that V_0^{T} is typically about $10 E_{\text{R}}$ for loop sizes at which the force response is peaked. This implies that

Fig. R1: **Insensitivity to lattice depth uncertainty.** (a) Lattice depth uncertainty tolerance V_0^T as a function of loop size Δq . (b) Interference fringe frequency versus loop size Δq .

if we want to reach a relative measurement accuracy of 10^{-6} , we only need to reduce σ_{V_0} below $0.01 E_R$, which is experimentally accessible.

In practice, the uncertainty σ_{V_0} is dependent on lattice depth calibration as well as lattice fluctuations. As presented in the Supplementary Information, the approach we took for lattice depth calibration is amplitude-modulation spectroscopy, which typically gives better precision than Kapitza-Dirac diffraction. The statistical uncertainty of the calibrated lattice depth at a given PID control voltage is about $0.2\% \sim 0.3\%$ with 95% confidence. The major systematic error for this calibration is from atom number counting and initial quasimomentum uncertainty. We believe the spectroscopic resolution of this approach can be further enhanced by reducing the detection noise and cooling the atomic cloud even more. On the other hand, lattice fluctuations contribute more significantly to σ_{V_0} . As shown in Fig. S6 of the Supplementary Information, the typical lattice depth drift is currently about $0.1 E_R$ shot-to-shot in our experiments. We believe this drift is mostly induced by laser pointing noise. With a more compact setup using fiber collimated lattice beams, the fluctuations can be greatly reduced.

In summary, the magic condition ensures that precision measurements are viable as long as the lattice depth uncertainty is moderately small. Such a condition is experimentally feasible after sufficient upgrades in lattice stabilization and calibration. The present work is a demonstration of a new approach to trapped-atom interferometry. More efforts regarding how to perform precision measurements are needed in future work, but are beyond the scope of this paper. To address the reviewer's concern, we have added in the Supplementary Information discussions about lattice depth uncertainty.

Supplementary Information lines 200-203 ...Scanning the center frequency of this sweep (Fig. S3b) produces a resonance peak, from which we can extract the band energy difference by fitting. The statistical error of characterizing the resonant frequency is typically 0.2% ~ 0.3% with 95% confidence. From the fitted band gap, we theoretically calculate the corresponding lattice depth...

Supplementary Information lines 311-341 We added a new section in the Supplementary Information to discuss lattice depth uncertainty.

Comment 1.2

Another concern relates to the spatial resolution of force measurements. As the authors emphasize, a longitudinal harmonic confinement prevents the two split wavepackets from recombining at the critical final quasimomentum at the same time, thus reducing the interference contrast. For this reason, any attempt to measure a spatially varying force would likely face the same limitation. For these two reasons, I conclude that, although I consider this work a remarkable piece of physics and a source of inspiration for future developments in the field, the technique still lacks a clear path toward practical applications in precision sensing.

Response: We agree that the fringe contrast is sensitive to spatial force inhomogeneity, and that a gradient in force can introduce an extra phase that is dependent on the initial position and velocity of the atomic cloud. We note that this is an issue shared by traditional freefall atom interferometers, for which inhomogeneity-induced contrast reduction and phase shifts have been extensively discussed, e.g., in [Peters *et al.*, 2001 *Metrologia* 38 25]. A scheme has been proposed and implemented to cancel this phase shift and recover perfect contrast [Amico *et al.*, PRL 119, 253201 (2017)]. While a complete resolution of this broader matter wave interferometry issue is well beyond the scope of the present manuscript, as the reviewer implies it will be a valuable target of future work to develop a protocol to deal with inhomogeneity within the interferometry framework we present. The flexibility of the interferometer design framework we present should enable a variety of approaches including some that are not possible in freefall. One simple palliative approach for example would be to optimize a loop for operation at higher lattice depths, reducing the Bloch oscillation amplitude and thereby lessening the effects of spatial inhomogeneity while maintaining force sensitivity with mid-loop holds. More complex approaches based on mid-loop momentum kicks may also be possible.

On the other hand, if the gradient in force is approximately constant over the spatial extent of the interferometer, it is possible to calculate the induced phase shift and subtract off its effect once the gradient is calibrated by an independent measurement. A more detailed discussion is presented in our response to Comment 1.7.

Comment 1.3

The authors claim that the interferometer is noise-tolerant. Can they quantify how sensitive the splitting and recombination processes are to lattice position fluctuations?

Response: The reviewer raises an interesting and relevant question regarding potential sensitivity of beamsplitter operations to lattice position fluctuations. In our protocol, atoms are initially loaded into the ground band of the lattice, and their motion is confined by the transverse trapping potential from the lattice itself. Under slow lattice position fluctuations, the atoms adiabatically follow the instantaneous lattice position, such that the splitting

and recombination processes would take place when the atomic clouds are spatially overlapped. Because the splitting/recombination processes are induced by amplitude modulation with a frequency much higher than the transverse trapping frequency, these processes are not significantly affected by transverse lattice motion. Very fast lattice position fluctuations (whose frequencies are comparable or higher than the transverse trap frequency) are passively suppressed because the lattice position noise is mainly produced by thermal or mechanical effects.

Comment 1.4

Moreover, I imagine that the interferometric phase remains affected by unwanted lattice accelerations.

Response: Longitudinal lattice fluctuations manifest as lattice phase noise in the high-frequency limit and thus effective force fluctuation, which should be common-mode for the interferometer arms at different points along the lattice. Longitudinal lattice acceleration in the low-frequency limit is precisely what the interferometer senses, by the Equivalence Principle. Such accelerations could be reduced by lattice phase stabilization. In contrast, transverse force fluctuations (arising from beam motion perpendicular to the lattice k -vector) could excite higher transverse modes and lead to imperfect recombination. In our Supplemental Information section 6, we mention stray transverse field gradients as a potential cause of imperfect loop closure; in response to this comment we have added lattice acceleration as an additional potential confounding factor grouped under the same general category of force fluctuation.

Supplemental Information lines 261-263 Likewise, transverse force fluctuation arising from unwanted lattice acceleration will by the same mechanism degrade the interferometer contrast; this in turn can be addressed with mode filtering via a resonant cavity.

Comment 1.5

The final imbalance is said to be insensitive to fluctuations of V_0 . What about fluctuations in δV , i.e., the modulation amplitude of the lattice?

Response: The modulation depth δV contributes to the interferometer fringe in two ways. First, it determines the (ideal) fringe contrast. A fluctuating modulation depth would modestly reduce the fringe contrast. Recall that the ideal contrast C depends on the LZ transition probability \mathcal{P} as $C = 4\mathcal{P}(1 - \mathcal{P})$. Since $\partial C/\partial \mathcal{P}|_{\mathcal{P}=0.5} = 0$, the uncertainty in \mathcal{P} (so as the uncertainty in δV) only has a second-order contribution to the uncertainty in C . As an example, a 10% peak-to-peak fluctuation in δV only leads to a 0.5% reduction in the contrast. Secondly, the change in the modulation depth leads to a shift in the Stokes phase. Because the Stokes phase is bounded as $\phi_{\text{Sto}} \in [-\pi/2, -\pi/4]$, the phase shift induced by a fluctuating δV is usually small. For example, a 10% peak-to-peak fluctuation in δV would result in a 0.02 rad shift in the Stokes phase. To achieve extreme high precision, users of this technique should carefully reduce the fluctuations in δV using, for example, an extra layer of laser power PID that stabilizes the laser power into the modulated AOM. We have added language discussing this point in response to the reviewer's comment.

Supplemental Information lines 342-352 We added discussions about fluctuating δV as a source of systematic error in our Supplementary Information.

Comment 1.6

Considering my previous comments, what kind of measurement could realistically be performed with this interferometer to go beyond the current state of the art?

Response: A measurement to go beyond the current state of the art using this interferometer concept would depend on taking advantage of the large spacetime area available in a compact geometry which is the distinguishing feature of our approach. Atoms could be held in position after the first beamsplitter to enclose a large spacetime area for better force sensitivity without increasing the size of the measuring instrument or coarsening the spatial resolution (see the discussion of our measurement of initial steps in this direction in Supplementary Information section 8). Of course like any such measurement this would depend on good control of systematics; we do not expect that this interferometer concept will replace freefall interferometers, but rather that it will provide complementary advantages and drawbacks. The appropriate measurement modality will depend on the quantity being measured; while a complete discussion of this topic would be its own paper, we do note here that fifth force searches often benefit from spatial resolution, which may make a continuously trapped approach appealing. We anticipate that future work on the interferometer concept we have presented and prototyped here will focus on enhancing the sensitivity and controlling dominant systematics. To more clearly present this viewpoint, we have modified the statement about precision measurements in the Discussion section.

Main text lines 373-377 Compact trapped interferometers suitable for weak force measurement may find application in fifth force searches or similar probes of physics beyond the standard model as a complementary approach to freefall atom interferometry.

Comment 1.7

An estimate of the achievable sensitivity and spatial resolution is missing. Could the authors provide quantitative estimates of the force sensitivity and its dependence on spatial resolution?

Response: We agree that characterizing force (acceleration) sensitivity is a critical step towards the demonstration of precision measurements. We discuss the theoretical limit on the sensitivity, then estimate the sensitivity we have achieved in our experiment.

We begin by considering the simplest case in which the acceleration being measured is uniform. Fundamentally, the sensitivity we can achieve is limited by the standard quantum limit (SQL) for uncorrelated atoms, which gives a phase estimation error bound of $\delta\phi_{\text{Int}}^{\text{SQL}} = 1/\mathcal{C}\sqrt{N}$, where N is the number of atoms and \mathcal{C} is the measured contrast. The single-shot SQL of the acceleration sensitivity is then given by

$$\frac{\delta a^{\text{SQL}}}{a} = \frac{\delta\phi_{\text{Int}}^{\text{SQL}}}{\phi_{\text{Int}}} = \frac{1}{\mathcal{C}\sqrt{N}} \frac{\mathcal{F}}{\hbar\mathcal{A}}. \quad (\text{R8})$$

If we assume the beamsplitters couple the two bands near the Brillouin zone's edge or center, the above equation can be approximated as

$$\frac{\delta a^{\text{SQL}}}{a} \approx \frac{1}{\mathcal{C}T\sqrt{N}(\bar{\Omega}_{\text{U}} - \bar{\Omega}_{\text{L}})}, \quad (\text{R9})$$

where T is the interrogation time and $\bar{\Omega}_{\text{U(L)}}$ represents the average band energy of the upper (lower) band divided

by \hbar . When $N = 10^5$, $\mathcal{F} = 0.2 Mg$, and the loop size $\Delta q = 15.8\hbar k_L$, the above SQL gives a 1.7×10^{-8} relative acceleration sensitivity with a single shot, assuming a perfect contrast. The corresponding interrogation time is 431 ms.

Next, we consider the effects of a spatially inhomogeneous force (acceleration). Intuitively, a spatially varying force can both induce a phase shift and reduce the contrast. To quantify these effects, we assume the total force can be written as $\mathcal{F} + M\gamma x$. In addition, we assume γ is small such that the Bloch Theorem is approximately valid and the dispersion relation $\tilde{E}_{n,q}$ still holds. The center-of-mass motions of the two separated clouds are governed by the following classical equations:

$$\frac{dq}{dt} = \mathcal{F} + M\gamma x, \quad (\text{R10})$$

$$\frac{dx}{dt} = \frac{d\tilde{E}}{dq}, \quad (\text{R11})$$

where $x(t)$ and $\tilde{E}(q)$ represent the center-of-mass trajectory and the Floquet-Bloch band energy of the upper or lower band atomic cloud, respectively. These are nonlinear equations, and we will attempt to solve them perturbatively. Ignoring γ , the unperturbed solution is

$$q^{(0)}(t) = q_r + \mathcal{F}t, \quad (\text{R12})$$

$$x^{(0)}(t) = x_r + \frac{\tilde{E}(q^{(0)}(t))}{\mathcal{F}}, \quad (\text{R13})$$

where q_r, x_r denote the initial quasimomentum and position, respectively. We assume $q(t)$ and $x(t)$ can be written as a perturbation series of γ ,

$$q(t) = q^{(0)}(t) + M\gamma q^{(1)}(t) + O(\gamma^2), \quad (\text{R14})$$

$$x(t) = x^{(0)}(t) + M\gamma x^{(1)}(t) + O(\gamma^2). \quad (\text{R15})$$

Plugging the above ansatz into Eqs. (R10) and (R11), we find

$$\frac{dq^{(1)}}{dt} = x^{(0)}(t), \quad (\text{R16})$$

$$\frac{dx^{(1)}}{dt} = q^{(1)}(t) \times \left. \frac{d^2\tilde{E}}{dq^2} \right|_{q=q^{(0)}(t)}. \quad (\text{R17})$$

Solving the above equations gives

$$q^{(1)}(t) = x_r t + \frac{1}{\mathcal{F}^2} \int_{q_r}^{q_r + \mathcal{F}t} dq \tilde{E}(q), \quad (\text{R18})$$

$$x^{(1)}(t) = \int_0^t dt' q^{(1)}(t') \times \left. \frac{d^2\tilde{E}}{dq^2} \right|_{q=q_r + \mathcal{F}t'}. \quad (\text{R19})$$

Eq. (R18) shows that the quasimomentum shift induced by the acceleration gradient γ is dependent on the dispersion relation $\tilde{E}(q)$, which implies that the two interferometer arms have different final quasimomenta. This differential quasimomentum shift leads to a contrast reduction that can be evaluated from the final quasimomentum overlap of the two atomic clouds. In the following, we consider an interferometry setup as mentioned before: $N = 10^5$, $\mathcal{F} = 0.2 Mg$, and $\Delta q = 15.8\hbar k_L$. We also assume the final quasimomentum distributions are Gaussian

Fig. R2: **Acceleration gradient induced effects.** (a) Fringe contrast as a function of γ , evaluated as the overlap of final quasimomentum distributions. Assuming $\mathcal{F} = 0.2 Mg$ and $\Delta q = 15.8\hbar k_L$. (b) Acceleration gradient induced phase shift, estimated from the change of the spacetime area under the same interferometry parameters.

functions with a $0.13 \hbar k_L$ full-width-half-maximum, same as the initial state. As shown in Fig. R2(a), the fringe contrast in the presence of γ is calculated as the overlap of two Gaussian distributions displaced by the differential quasimomentum shift. We notice that γ needs to be as large as 5 s^{-2} to reduce the contrast by a half. Because it is experimentally straightforward to control γ below 1 s^{-2} (at which the contrast is only reduced to 0.88), we believe the gradient-induced contrast reduction is less relevant to a consideration of force sensitivity.

Since the interferometer phase scales with the enclosed spacetime area, we estimate the gradient-induced phase shift $\Delta\phi_{\text{Grad}}$ from the change of the spacetime area given by integrating Eq. (R19). For simplicity, we assume $x_r = 0$, which implies the force reference position is where the atomic cloud is split. Fig. R2(b) shows that when $\gamma = 3 \times 10^{-6} \text{ s}^{-2}$ (about the Earth’s gravity gradient), the induced relative phase shift is about 10^{-8} , which is comparable to the SQL of the phase sensitivity. This phase shift is a systematic error that can be treated if γ is measured independently. Indeed, in relevant work of freefall atom interferometry (Peters *et al.*, 2001 Metrologia 38 25), the authors obtained an absolute measurement of gravity after the correction from the gravity gradient measured by LaCoste-Romberg spring-type gravimeters.

Finally, we can estimate the acceleration sensitivity from the phase measurement in Fig. 4f in the main text. The fit gives a statistical error of 0.019 rad (68% confidence interval) for the interferometer phase, after a 20 min phase measurement. This corresponds to a 1.01×10^{-4} relative acceleration sensitivity under the experimental parameters $T_B = 10.56 \text{ ms}$ and $\Delta q = 0.4\hbar k_L$. The sensitivity could be enhanced by enlarging Δq , which would require improved control of transverse atomic motion.

In summary, the SQL of the relative acceleration sensitivity is $\mathcal{F}/\mathcal{C}\sqrt{N}\hbar\mathcal{A}$, which is comparable to traditional atom interferometers under the same interrogation time. The acceleration gradient γ weakly contributes to contrast reduction, but it induces a phase shift as a systematic error which requires careful handling. It will be our future work to improve the sensitivity in experiments. In response to the reviewer’s comment, we have included discussions about the SQL and the effects of force inhomogeneity in the Supplementary Information.

Supplementary Information lines 291-307 We added discussions about the SQL of the acceleration sensitivity in the Supplementary Information.

Supplementary Information lines 353-384 We added discussions about the effects of the acceleration gradient in the Supplementary Information.

Comment 1.8

Another potential source of contrast reduction could be different dispersions of the two wavepackets. Is this effect relevant, or do the two wavepackets perfectly match in width at the moment of the second beam splitter?

Response: Differential wavepacket spreading due to different band dispersion is indeed a noticeable effect, as can be seen in Fig. 1d; the finite-time traversal of the Landau-Zener crossing along with the differential group velocities of the two bands serves to spread out the higher-energy population in real space. However, this same mechanism is experienced in reverse at the second beamsplitter, which serves to re-compress the fraction returned to the P band, causing the two wavepackets to again match in width. Moreover, since the force translates the quasimomentum uniformly $q(t) = q(0) + \mathcal{F}t$ and preserves the quasimomentum distribution, the wavepacket width will depend to first order only on instantaneous quasimomentum and not on its history. In addition, since the bands are reflection-symmetric across the Brillouin zone, wavepacket width is an even function of quasimomentum. Thus, since our interferometric loop and beamsplitters are symmetrically placed in the Brillouin zone, the relative wavepacket widths should be identical at each beamsplitter. Any differential spreading experienced during the loop traversal will be reversed when the two wavepackets recombine. We have clarified this point in the main text.

Main text lines 118-122 ...as shown in Fig. 1c-d. Although the two wavepackets experience different dispersions, any differential spreading experienced during the loop traversal will be reversed when they recombine. Components of the atomic wavefunction which are separated at an initial Landau-Zener beamsplitter...

Comment 1.9

I am wondering how condensed the atomic cloud used in the experiment is. Could the interferometric scheme also work with a thermal sample? Does the interferometer have a white-light-like character?

Response: Regarding the first question, we achieve condensate fractions greater than 95% (in fact with no detectable thermal fraction) before loading into the lattice, so the cloud is in the deeply quantum degenerate limit for the experiments we report. We've added text about this in the methods section.

Main text lines 418-421 The condensate fraction is greater than 95%, having evaporated in the trap until the thermal components are no longer visible.

We believe that the experiment could in principle also work with a thermal ensemble, as the physics is fundamentally based on a single atom picture, though it may be more challenging to extract a good signal. The scheme requires the atoms to be in a single band, so this limits us to a temperature that is not large compared to the lattice recoil temperature $1.2 \mu\text{K}$. The temp also needs to be small enough that the initial momentum distribution is sufficiently far away from the coupling quasimomenta to ensure adiabaticity and maintain equal population splitting. We have added some discussion on this in the Supplementary Information.

For all quasimomentum components, because the atoms are Bloch oscillating they see the same evolution in quasimomentum under a force and thus accumulate the same relative phase once they are in a superposition of two bands, giving the same output fraction. This should remain true for a thermal population.

Supplementary Information lines 102-106 We assume a coherent sample for this experiment and discussion, but we believe this model should hold for a thermal ensemble as well if the temperature is low enough that the initial momentum distribution is mostly confined to a single band. In the absence of interaction it should be sufficient that the temperature be this cold only along the lattice axis, which could be achieved with a velocity selective filtering process, as long as the transverse confinement is wide and deep enough that all atoms see about the same lattice depth.

Comment 1.10

Considering that lithium is a very light species, what would significantly change if this scheme were implemented using heavier atoms, in terms of optical power and final sensitivity?

Response: In addition to the straightforward change in the amplitude of Bloch oscillations, the most significant consequence of using heavier atoms will be the increased possibility of LZ tunneling to other bands. In our scheme, a key limiting process will be LZ tunneling from the D band to the F band at the Brillouin zone edge; such tunneling represents a “leak” which removes atoms from the interferometer loop. Here we discuss how this process depends on atomic mass.

We assume the lattice laser wavelength does not change when using different species. With a larger atomic mass, the recoil energy E_R becomes smaller, rescaling the static band structure. Therefore, the proportionality V_0^M/E_R remains constant for a given loop size. We can define the normalized magic depth as $\tilde{V}_0^M = V_0^M/E_R$. In the following, we assume the interferometer operates at the fixed \tilde{V}_0^M for a given loop size when we consider different atomic species. The LZ theory tells us that the transition probability through a Bloch band gap is given by

$$\mathcal{P}_{\text{Bloch}} = e^{-1/\kappa}, \quad \kappa = \frac{8}{\pi^2} \frac{\hbar\omega_B/E_R}{(\Delta/E_R)^2}, \quad (\text{R20})$$

where Δ is the band gap between D and F and $\omega_B = 2\pi/T_B$ is the Bloch frequency. For ${}^7\text{Li}$, the above equation gives $\mathcal{P}_{\text{Bloch}} = 6.2 \times 10^{-11}$ through the D - F gap at the Brillouin zone edge, when $V_0 = 8.45 E_R$ and $T_B = 10.7$ ms. Notice that the band gap scales with E_R , so that Δ/E_R remains constant. Therefore, $\kappa \propto M^2 a$ where a is the acceleration. This tells us that when we operate the interferometer under the magic condition, the range of acceleration we can measure scales with $1/M^2$ if we want to maintain a low coupling rate to unwanted bands. Fortunately, this problem is not in principle unsolvable. In a manner similar to the measurements shown in Fig 4c and d, we can superimpose another modulation frequency that couples the D band to the S band near the Brillouin zone edge, such that D band atoms are shelved in S when they are close to $q = k_L$, avoiding the tunneling leakage and allowing the use of heavier atoms. This scheme will of course modify the interferometer phase and require a somewhat different magic condition.

Because the recoil energy E_R is smaller for heavier atoms, the amount of optical power required under the magic condition is smaller if we assume a fixed laser detuning and atomic dipole matrix element. Meanwhile, the SQL of the acceleration sensitivity given by Eq. (R9) scales with M for a fixed interrogation time T . Equivalently, it takes a longer interrogation time for heavier atoms to reach the same sensitivity as lighter atoms.

Supplementary Information lines 404-426 We added a new section in the Supplementary Information to discuss the effects of using heavier atoms.

2 Response to reviewer 2

General Comments

The manuscript describes the theory and experimental implementation of an optical lattice atom interferometer that uses amplitude modulation of the optical lattice to perform the atom optics operations: beam splitter, mirror, recombination. The operations are achieved by transferring Li atoms between the p and d bands of the optical lattice, taking advantage of "magic trap depths", specific laser intensities where the interferometer is shown to be first order insensitive to lattice amplitude noise. The scheme presented is in the same vein as several similar implementations using frequency modulation by the JILA and U Washington teams, but different in important ways.

It remains to be seen whether experimental sensitivity and systematic suppression can reach level competitive with the current state of the art. Despite this, the results are an important step forward for lattice atom interferometer and highlight a new type of optical lattice atom interferometer. The experimental data is compelling and interpretation is physically sound. The manuscript is well written, and the data is nicely presented.

Response: We thank the referee for their thoughtful comments and suggestions. We appreciate their characterization of the manuscript as an important step forward, the experimental results as compelling, and the interpretation as physically sound. We respond to all the points and suggestions of the referee below, and have made changes to the manuscript accordingly.

Comment 2.1

The interferometer scheme leads to an unusual interferometer phase dependence on the parameter of interest, i.e it is proportional to $1/\text{force}$, instead of the usual proportionality to force. This means that keeping the force small is important for achieving a large phase and therefore good sensitivity. This is acknowledged by the authors in the conclusion, "The force sensitivity in the weak-force regime improves as $1/F$, a feature that could be exploited in more generality by introducing an accelerated lattice to realize a frame transformation and cancel part of the force." Of course, the problem is that the main advantage of the interferometer, ability to achieve long measurement times, disappears, since the lattice has to be traveling at g to cancel out the force. I wonder if this can be commented on.

Response: We agree with the referee's point here. We have deleted this sentence to avoid confusion.

Main text lines 369-374 ...and intrinsic robustness against trap-induced dephasing and pulse errors. ~~The force sensitivity in the weak-force regime improves as $1/\mathcal{F}$ is a feature that could be exploited in more generality by introducing an accelerated lattice to realize a frame transformation and cancel part of the force.~~ Compact trapped interferometers...

Comment 2.2

The sensitivity of the interferometer seems to be far from competitive for precision measurements. However, this is an important consideration on whether it would be useful for applications. The interferometer is phase stable, but a short discussion of possible systematic effects could help the reader assess possible applications. This is particularly the case given the complex residual effects from interactions with the optical lattice.

Response: We agree with the reviewer that the sensitivity we have attained in this first demonstration of the technique is far from state-of-the-art. A discussion of the ultimate theoretical sensitivity is presented in our response to Comment 1.7 above, which shows the single-shot standard quantum limit (SQL) of an acceleration measurement:

$$\frac{\delta a^{\text{SQL}}}{a} \approx \frac{1}{\mathcal{C}T\sqrt{N}(\bar{\Omega}_{\text{U}} - \bar{\Omega}_{\text{L}})}, \quad (\text{R21})$$

where \mathcal{C} is the measured fringe contrast; T is the interrogation time; N is the atom number; $\bar{\Omega}_{\text{U(L)}}$ represents the average band energy of the upper (lower) band divided by \hbar . Our experiments have demonstrated a relative phase sensitivity of 1.01×10^{-4} with a 20 min measurement. Improving the sensitivity requires enlarging the interferometer loop size Δq , which is limited by the following systematic effects and noise:

- Lattice depth uncertainty. As mentioned in our response to comment 1.1 above, the lattice depth uncertainty is a key systematic error for trapped interferometers; in the interferometer design we present, its effects can be mitigated by the magic band structure.
- Modulation depth. Fluctuations in δV can lead to a reduced contrast and a shift in Stokes phase, as discussed in our response to comment 1.5 above.
- Acceleration gradient. If the acceleration being measured is not uniform, contrast reduction and phase shift can be induced, as discussed in our response to comment 1.2 above.
- Residual interaction. The Feshbach magnetic field is calibrated using atomic microwave spectroscopy with a precision of about 100 mG, which corresponds to a residual scattering length of $0.007 a_0$. This residual interaction can cause a 9 mHz mean-field energy shift which is negligible in our experiments.
- Detection noise. Since we use absorption imaging to determine the atom number, the background photon shot noise could dominate quantum projection noise.
- Field gradient noise. The force that our interferometer is sensitive to is produced by magnetic coils. Although the current flowing into the coils is PID controlled, the residual noise of 2×10^{-4} can potentially limit how far we can push the sensitivity.
- Transverse motion. As discussed in the Supplementary Information section 6, we believe the transverse motions of the atoms are the major source of contrast reduction. These motions can be induced by the transverse magnetic field curvature or the transverse jitter of the lattice beam.

In response to the reviewer comments we have added detailed analyses of these matters in Supplementary Information section 10.

Supplementary Information lines 308-403 We have added a new section in the Supplementary Information to discuss the possible systematics and noises.

Comment 2.3

“The most likely candidate for the reduction of fringe contrast at higher is an inhomogeneous transverse field gradient, causing imperfect loop closure” - Using atoms in a magnetically sensitive state makes the experiment quite sensitive to environmental fields. Are collisions impossible to deal with in a magnetically insensitive state?

Response: We agree that interactions can make it difficult to work with more magnetically insensitive species, and that is a consideration for future experiments using other isotopes. In addition to isotope selection, one can use large lattice beam widths and work at reduced densities. For example ^{88}Sr is an appealing possibility due to its weak background scattering length of about $-1 a_0$, though the heavier atom brings its own set of challenges. Alternatively, field noise can be reduced, by shielding or active stabilization with feedforward, which can minimize the impact of field noise on the experiment down to 0.1 mG (see, e.g., *PRL* **135**, 133401 (2025)).

Comment 2.4

“The interferometric sensitivity is set by the spacetime area enclosed by the interferometer loop and thus scales quadratically with freefall time for untrapped particles.” - I suggest replacing "set by" with "proportional", since sensitivity also depends on measurement rate – duty cycle and number of atoms per cycle.

Response: We agree that “proportional to” is a more precise phrasing and have edited this sentence accordingly.

Main text lines 22-25

The interferometric sensitivity is set by spacetime area enclosed by the interferometer loop and thus scales quadratically with freefall time for untrapped particles.

→

The interferometric sensitivity is **proportional to** spacetime area enclosed by the interferometer loop and thus scales quadratically with freefall time for untrapped particles.

Comment 2.5

“without requiring mode filtering with a resonant cavity.” Cavity wouldn’t filter the amplitude noise unless the noise is at very high bandwidth, beyond the resonance linewidth of the cavity. This seems unlikely since most laser intensity noise is in the acoustic band.

Response: We thank the reviewer for pointing out this ambiguity. Our intention in this sentence was not to refer to amplitude noise filtering, but rather to reducing distortions of the spatial mode (i.e. the transverse intensity profile), which can cause trap inhomogeneities. We agree that a cavity would not effectively filter relevant amplitude noise. We believe the clause is unnecessary and have removed it.

Main text lines 56-58 We experimentally verify that magic Floquet-Bloch bands exhibit first-order insensitivity to lattice amplitude noise.

Comment 2.6

In figure 2b, it would make sense to convert from imbalance to phase. That would make the assessment of intensity insensitivity more accurate, which is now dependent on being on the side or extrema of the fringe.

Response: We would like to clarify that we do not use the experimental results presented in Figure 2b in the main text to determine our magic condition; we calculate it numerically from first principles (Figure 2a), so the assessment of intensity insensitivity does not depend on measurement. Rather, we present the results of Figures 2b-d to confirm our numerical predictions. Consequently, the assessment of the magic condition does not depend on the magic lattice depth appearing at the extrema of the fringe for the particular force we happened to choose. In Figure 2d we scan an entire period of the fringe at the three depths close to the magic condition and see that their overlap is independent of being at the precise force that happens to produce an extremum in the fringe.

We understand the referee's point about imbalance versus phase, and considered making this change. The issue is that since the fringe is periodic with respect to Bloch period and inverse force, a mapping from interferometer phase to imbalance is not injective, so inverting that map would require extra assumptions. For these reasons, we feel that the experimentally measured imbalance is still the most unambiguous presentation of the measurements which confirm of the predicted magic condition.

3 Response to reviewer 3

General Comments

In this work, Chai et al. present a flexible method of precision measurement for measuring extremely weak applied forces. This method of interferometry is described through the lens of Bloch oscillations via Floquet physics and Landau-Zener transition probabilities. Unlike many other types of interferometers, this new class of interferometer is extremely insensitive to many external factors and can be achieved in trap. In reviewing this paper, I was impressed by how thorough the authors were in their description of the new class of interferometer and was struck by the cleanliness of the data presented in the paper. In a time when new methods for quantum sensing are highly sought after, I find this work to be well-timed and highly beneficial to the field of precision measurements. I highly recommend this work for publication in Nature Communications, given the above comments and the relevant nature of the work to a growing field of quantum science and technology. I firmly believe that this work, along with the Supplemental Information, will be a relevant guide to those who would like to implement similar experimental methods.

Response: We are grateful to the reviewer for their thoughtful engagement with the manuscript. We appreciate their characterization of the work as well-timed and highly beneficial to the field, and their recommendation that it be published in Nature Communications.

Comment 3.1

One general overarching note is that, while the Supplementary Information is very thorough, it seemed to be that the supplementary information was needed in certain parts of the main manuscript to understand what was going on. The authors may want to consider small changes to the main manuscript to make sure that the Supplementary Information is providing additional information but is not needed for one to understand what is going on in the main text.

Response: We thank the reviewer for this useful feedback on the presentation of our work, and we have made changes in response which we think improve the manuscript. In the spirit of better explaining our admittedly complex experiment, we focused our efforts in editing the “Working Principle” section, whose references and connections to Supplemental Information are most relevant to a reader’s understanding (other direct references to Supplemental Information in the rest of the text tend to describe calibration procedures or similar, more typical digressions from the main narrative). We list our changes here:

- Our introduction of Floquet-Bloch band structures certainly elided a great deal of explanation left to the Supplemental Information. Indeed, in one sentence, we introduce in quick succession quasimomentum, resonant quasimomentum, band index (and our notation thereof), and the resonant condition, in the order in which they appear in an equation. We have reordered this sentence to improve clarity:

Main text lines 76-87

As shown in Fig. 1c, the modulation opens up gaps between Floquet-Bloch quasienergy bands when the resonance condition $E_{n,q_r} - E_{n',q_r} = \hbar\omega$ is satisfied, where $E_{n,q}$, $E_{n',q}$ are the bare Bloch band energies for bands labeled n and n' (henceforth S, P, D , etc., following orbital notation), q is the quasimomentum, q_r is the resonant quasimomentum, and \hbar is the reduced Planck constant.

→

Just as quasimomentum q arises from the breaking of continuous to discrete space translation symmetry, the analogous breaking of continuous to discrete time translation symmetry gives rise to *quasienergy* bands, which are periodic in $\hbar\omega$, where \hbar is the reduced Planck constant. As shown in Fig. 1c, the modulation opens up gaps between Floquet-Bloch quasienergy bands where they intersect, at quasimomentum $q = q_r$. For band indices n and n' (henceforth S, P, D , etc., following orbital notation) of the two bare Bloch bands $E_{n,q}$, $E_{n',q}$, the resonance condition is $E_{n,q_r} - E_{n',q_r} = \hbar\omega$.

- The explanation of how amplitude modulation δV controls the Landau-Zener tunneling probability was previously omitted. We have slightly expanded the discussion of this point:

Main text lines 90-92

By tuning the modulation depth δV , the Landau-Zener transition probability p can be adjusted to 50% [34].

→

Tuning the modulation depth δV controls the gap, and thus the Landau-Zener transition probability \mathcal{P} can be adjusted to 50% [34].

- The presentation of imbalance, the main experimental output of our interferometer, likewise skipped much of its derivation and presents many new symbols in quick succession. We also recognize that the ideal contrast equation can be a topic left to Supplemental Information. This sentence has been rewritten:

Main text lines 125-134

This phase is measured by interfering the two pathways at the second beamsplitter, forming a Landau-Zener-Stückelberg-Majorana interferometer [38, 39] and resulting in a final population imbalance between the two output bands (Supplementary Information section 1) $\mathcal{I} = p_L - p_U \approx C \cos \phi_{\text{Int}}$, where $L(U)$ denotes the lower (upper) Floquet-Bloch band, $p_{L(U)}$ the corresponding population fractions, and the ideal contrast $C = 4p(1 - p)$.

→

Recombination of the matter waves at the second beamsplitter completes a Landau-Zener-Stückelberg-Majorana interferometer [38, 39]. The differential phase acquired by each arm ϕ_{Int} manifests as a population imbalance between the two output bands $\mathcal{I} = p_L - p_U \approx C \cos \phi_{\text{Int}}$ by the adiabatic-impulse approximation [40], where $p_{L(U)}$ denotes population fraction in the lower (upper) Floquet-Bloch output band. The contrast C is maximized by 50-50 beamsplitting fraction $\mathcal{P} = 0.5$.

- We reference Methods parenthetically thrice in the Experimental Implementation section. To remove unnecessary distractions from the narrative, we have consolidated these into a single reference at the end of the section:

Main text lines 143-145

The experimental apparatus with which we realize this interferometer builds upon previous work (see [23] and Methods).

→

The experimental apparatus with which we realize this interferometer builds upon previous work [23].

Main text lines 158-160

A strong 120 kHz lattice amplitude modulation pulse is then applied to transfer the atoms from the S to P band with 100% Landau-Zener probability (see Methods).

→

A strong 120 kHz lattice amplitude modulation pulse is then applied to transfer the atoms from the S to P band with 100% Landau-Zener probability.

Main text lines 178-180

Finally, the modulation is ramped down in 100 μ s to convert atoms in upper (lower) Floquet-Bloch bands back into the static P (D) Bloch bands. We perform band-mapping [41] and absorption imaging to read out the band populations with high fidelity (see Fig. 1e and Methods).

→

Finally, the modulation is ramped down in 100 μ s to convert atoms in upper (lower) Floquet-Bloch bands back into the static P (D) Bloch bands. We perform band-mapping [41] and absorption imaging to read out the band populations with high fidelity (Fig. 1e).

Main text line 186 See Methods for details.

Comment 3.2

In Figure 1e, it would be nice to note the imaging time-of-flight time and to provide a scale on the image to give an idea for the relative size of the signals and their separation in the experiment. The time-of-flight should also be included in the relevant part of the methods section.

Response: We have added a scale bar to Fig. 1e and modified the text accordingly to mention the TOF time. While adding the scale bar to Fig. 1e, we noticed an error in the determination of pixel size. We were using the camera pixel size without including the imaging magnification of 3.3. We have corrected the spatial axes of Fig. 1 and Fig. S8 accordingly. This error only influences the axes of these plots and does not affect any other measured quantities.

Main text Fig. 1 caption ...(e) Band-mapping spatially separates the two output port populations after a 4 ms time of flight...

Main text Fig. 1e Added a scale bar.

Main text Fig. 1c, 1d, Supplementary Information Fig. S8 Corrected the spatial scales.

Main text lines 458-459 After band mapping with a 4 ms time of flight (Fig. 1e), the interferometer output...

Comment 3.3

On the 3rd page, first column, in the sentence that starts with “This phase is measured by interfering...” (lines 119-127), the authors define the upper and lower populations, but then in the ideal contrast equation, p is not defined. I understand that this is defined in the Supplementary information, however, this parameter should at least be named in the main text, e.g. “... where p is the Landau-Zener probability (see Supplementary Information section 1).” The authors could then move the reference to Supplementary Information to the end of this sentence, rather than the start of the sentence. The main reason I found this to be confusing is that the same letter was used for population fractions p_U and p_L , so having a parameter that is p , sans subscript, led me to become confused that maybe this was another population fraction term.

Response: We agree with the reviewer that the definition of the Landau-Zener probability could be presented more clearly. In order to address this and avoid confusion about the ideal contrast equation, we have changed its symbol to \mathcal{P} and made adjustments to the manuscript accordingly. This change was taken into consideration in the above response (Lines 125-134 in the main text), where we migrate information between the main text and supplementary information for additional clarity.

Main text lines 90-92

By tuning the modulation depth δV , the Landau-Zener transition probability p can be adjusted to 50% [34].

→

Tuning the modulation depth \$\delta V\$ controls the gap, and thus the Landau-Zener transition probability \mathcal{P} can be adjusted to 50% [34].

Main text lines 125-134

The differential phase acquired by each arm ϕ_{Int} manifests as a population imbalance between the two output bands $\mathcal{I} = p_L - p_U \approx C \cos \phi_{\text{Int}}$ by the adiabatic-impulse approximation [40], where $p_{L(U)}$ denotes population fraction in the lower (upper) Floquet-Bloch output band. The contrast C is maximized by 50-50 beamsplitting fraction $\mathcal{P} = 0.5$.

Supplemental Information Eq. (S25)

$$|\psi(0)\rangle = \int_{-\hbar k_L}^{\hbar k_L} dq_0 \mathcal{P}(q_0) |\tilde{\varphi}_{l,q_0}(0)\rangle,$$

$$\rightarrow$$

$$|\psi(0)\rangle = \int_{-\hbar k_L}^{\hbar k_L} dq_0 \mathcal{P}(q_0) |\tilde{\varphi}_{l,q_0}(0)\rangle,$$

Supplemental Information line 96

where $\mathcal{P}(q_0)$ is the quasimomentum distribution function.

\rightarrow

where $\mathcal{P}(q_0)$ is the quasimomentum distribution function.

Supplemental Information Eq. (S18)

$$\mathcal{I} = p_L - p_U \approx 4p(1-p) \cos \phi_{\text{Int}}$$

\rightarrow

$$\mathcal{I} = p_L - p_U \approx 4\mathcal{P}(1-\mathcal{P}) \cos \phi_{\text{Int}}$$

Supplemental Information Eq. (S19)

$$p = e^{-2\pi\delta}$$

\rightarrow

$$\mathcal{P} = e^{-2\pi\delta}$$

Supplemental Information lines 218-222

We empirically select the modulation depth for which the output populations are equal, ensuring that both beamsplitters in the full interferometer sequence set the Landau-Zener transition probability $p = \exp(-2\pi\delta) = 0.5$.

However, we only calibrate this modulation depth for the magic condition; since we fix δV , the beamsplitting fraction P changes slightly for other lattice depths as a result of changes to $|\varphi_{n,q_r}\rangle$ (see Eq. (S12)).

\rightarrow

We empirically select the modulation depth for which the output populations are equal, ensuring that both beamsplitters in the full interferometer sequence set the Landau-Zener transition probability $\mathcal{P} = \exp(-2\pi\delta) = 0.5$.

However, we only calibrate this modulation depth for the magic condition; since we fix δV , the beamsplitting fraction \mathcal{P} changes slightly for other lattice depths as a result of changes to $|\varphi_{n,q_r}\rangle$ (see Eq. (S12)).

Comment 3.4

The authors calibrated the modulation amplitude δV only at the magic condition (Supp Mat section 4.3). How sensitive are the results to small deviations in δV across the lattice depths used in the experiment? When plotting the imbalance as a function of lattice depth (e.g. in Fig. 2b), it might be helpful for the authors to provide a brief comment on how possible uncertainties in δV could affect these measurements.

Response: We thank the reviewers for raising this important question. Deviations in δV chiefly affect the contrast of the interferometric fringe as we note in Supplemental Information section 4.3, since it sets the band gap and thus the Landau-Zener tunneling probability \mathcal{P} . In our response to the first reviewer, we noted that the reductions in fringe contrast near $\mathcal{P} = 0.5$ scale quadratically with deviations in \mathcal{P} . In turn, the fluctuations in δV contribute quadratically to contrast reduction. A quick calculation shows that near $\mathcal{P} = 0.5$, a 10% peak-to-peak fluctuation in δV reduces contrast by only 0.5%.

δV also enters the definition of the Stokes phase (Eq. (S23)) via the adiabaticity parameter. However, as we show in Fig. S2, the Stokes phase is bounded between $[-\pi/4, -\pi/2]$ and monotonic, while the dynamical phase varies by a factor of roughly 2500 more within the same range of Bloch periods; thus, fluctuations in δV should have negligible effect on the measurement of the interferometric phase in Fig. 2b. A quick calculation gives a 0.02 rad phase deviation under a 10% peak-to-peak fluctuation in δV .

The observed reduction in contrast across the range of lattice depths in Fig. 2b is likely instead a result of the complicated dependence of the band gap Δ_{q_r} (Eq. (S12)) on the lattice depth V_0 via the Bloch eigenstates $\varphi_{n,q}$ (Eqs. (S3) and (S4)) and the coupling quasimomentum q_r , which for a fixed modulation frequency will change with V_0 . This could in principle be compensated by calibrating the modulation amplitude for every lattice depth, but since we only intend to illustrate the insensitivity of the interferometric phase in Fig. 2b rather than the contrast, such repeated calibration would be beyond the scope of the figure. In addition, we observe noisier measurements of imbalance far from the magic condition (e.g. Fig. 2c), which at the extrema of fringes can only reduce imbalance and thus perceived contrast; this further highlights the importance of the magic condition.

Supplemental Information lines 342-352 We added discussions about fluctuating δV as a source of systematic error in our Supplementary Information.

Comment 3.5

In a literature review for similar work using Floquet-state engineering for interferometry, we found the following paper relevant to this work: T. Rodzinka et al, Nature Comm 15, 10281 (2024). While this paper does not take away from the novelty of the current work under review, I found that it may be important to cite this work, perhaps in the Methods section, to make it clear that this work stands alone from other work previously done with these types of methods. For example, “Since the lattice is oriented horizontally along the x -axis, gravity does not produce a force along the lattice, as is the case in Ref. \cite{Rodzinka2024}.” Of course, I can’t recall if the limit on number of references for Nature Comm is 50, but considering the similar Bloch oscillation method used in that paper, we thought it would be good to mention prior to publication.

Response:

We thank the reviewer for pointing out this missing citation. We have included it as suggested.

Main text lines 429-431 We have added a citation in methods referencing T. Rodzinka’s 2024 paper. "Since the lattice is oriented horizontally along the x -axis, gravity does not produce a force along the lattice, as is the case in Ref. [52]."

Comment 3.6

We are assuming that the “shallow Feshbach zero-crossing” for Lithium is sufficiently shallow that when the magnetic gradient is applied, that the change in scattering length along the extent of the BEC induces negligible interaction-induced dephasing. Is this a correct assumption of the experimental setup? In the main manuscript and in the supplemental information, I did not get a sense of the relative strength of the magnetic field gradient used in the experiment. In section 5 of the supplemental information, under **Gradient Coil Current**, it may be nice for the authors to give a range for the gradients using in the work, in relatable units of G/cm.

Response: We agree with the reviewer that the change in scattering length is negligible. Based on our force calibration, the typical magnetic field gradient along the lattice axis is 1.25 G/cm. At the zero-crossing (543.6 G), the slope of the scattering length versus the magnetic field is $0.071 a_0/\text{G}$, where a_0 is the Bohr radius. In our experiments, the atomic position deviation from where we calibrated the zero-crossing is about 0.5 mm, which maximally leads to a scattering length of $a_s \pm 0.004 a_0$ induced by the field gradient. Considering that the typical atomic density is $n_{\text{atom}} = 1.4 \times 10^{18} \text{ m}^{-3}$, the induced mean-field energy shift will be $4\pi\hbar^2 a_s n_{\text{atom}}/M \approx 5 \text{ mHz}$ per particle. This is a very small shift on the time scales of our experiment.

We have also added the suggested note regarding the typical size of the gradient in the place the reviewer suggests.

Supplemental Information lines 385-394 We added relevant discussions in section 10.4 of the Supplementary Information.

Comment 3.7

In Figure 4, the x -axes are given in terms of the Bloch period T_B ms or the quasimomentum q ($\hbar k_L$). Because these two quantities are directly related through the equation $q(t) = q(0) + \mathcal{F}t$, the figure could be made clearer if the authors added a small annotation or a visual cue to remind the readers of this mapping.

Response: We thank the referees for this helpful suggestion in clarifying a complicated figure. We agree that we could take this opportunity to remind readers of the mapping between quasimomentum in Figs. 4(a-c) and T_B in Figs. 4(d-f). We have included such an annotation in the caption of Figure 4.

Main text Figure 4 caption (a) Dressed energies of the $\Delta q = \hbar k_L$ loop and the corresponding modulation waveform. The modulation is pulsed to avoid coupling with the F band. Recall that quasimomentum evolves via the acceleration theorem as $q(t) = q(0) + 2\hbar k_L (t/T_B)$, so the time to traverse an interferometer loop of fixed quasimomentum separation is proportional to T_B .

Comment 3.8

In the Supplementary Information, lines 62-64 state, “We adopt a coarse-graining argument [3], which states that an infinite number of avoided crossings below a certain scale can be ignored because of the finite experimental time scale.” In this work, how did the authors choose the scale for the neglected avoided crossings? How is the threshold quantitatively justified or verified in their analysis?

Response: A more quantitative analysis can be given by the Landau-Zener formula which tells the transition probability through a Floquet band gap,

$$\mathcal{P} = e^{-2\pi\delta}, \delta = \frac{\Delta^2}{4\hbar v}, \quad (\text{R22})$$

where Δ is the Floquet band gap size and v is the Landau-Zener sweep velocity (see also Eq. (S19) in the Supplementary Information). Avoided crossings can be regarded as negligible when the induced LZ tunneling probability \mathcal{P} is approximately 100%. Using the parameters in Fig. 1 in the main text, the typical beamsplitting gap size in our experiments is about 1.53 kHz which leads to $\mathcal{P} = 50\%$. The next largest gap within the interferometer loop is between S and D , which is induced by the "two-photon" transition at $q = \pm 0.88 \hbar k_L$. This gap has a size of 12.4 Hz, and the corresponding tunneling probability is $\mathcal{P} = 99.994\%$ which justifies that it is negligible. All the other Floquet gaps within the interferometer loop are smaller than 12.4 Hz, therefore we can safely ignore them and apply the instantaneous-Floquet-state formalism.

Comment 3.9

In Eq. (S43) of the Supplementary Information, the authors approximate the change in Stokes phase as a function of the change in lattice depth V_0 is very small compared to the change in the Dynamical phase over the same range. It may be relevant to make a reference to both Eq. (S23) and Figure S2 preceding or following this approximation.

Response: We have added references to Eq. (S23) and Figure S2 to the lines immediately preceding Eq. (S43).

Supplemental Information lines 158-159

Since the Stokes phase is monotonic in δ and bounded by $[-\pi/2, -\pi/4]$, it varies far less than the dynamical phase for the same range of force and lattice depth.

→

Since the Stokes phase (Eq. (S23)) is monotonic in δ and bounded by $[-\pi/2, -\pi/4]$, it varies far less than the dynamical phase for the same range of force and lattice depth, as shown in Fig. 2.